# THE PRICE OF AMORTIZED INFERENCE IN SPARSE AUTOENCODERS

**Wenjie Sun[1], Di Wang[2], Lijie Hu[1,*]**

[1]Mohamed bin Zayed University of Artificial Intelligence (MBZUAI)
[2]King Abdullah University of Science and Technology (KAUST)
{wenjie.sun,lijie.hu}@mbzuai.ac.ae, di.wang@kaust.edu.sa

## ABSTRACT

Polysemy has long been a major challenge in Mechanistic Interpretability (MI), with Sparse Autoencoders (SAEs) emerging as a promising solution. SAEs employ a shared encoder to map inputs to sparse codes, thereby amortizing inference costs across all instances. However, this parameter-sharing paradigm inherently conflicts with the MI community's emphasis on instance-level optimality, including the consistency and stitchability of monosemantic features. We first reveal the trade-off relationships among various pathological phenomena, including feature absorption, feature splitting, dead latents, and dense latents under global reconstruction-sparsity constraints from the perspective of training dynamics, finding that increased sparsity typically exacerbates multiple pathological phenomena, and attribute this trade-off relationship to amortized inference. By reducing reliance on amortized inference through the introduction of semi-amortized and non-amortized approaches, we observed that various pathological indicators were significantly mitigated, thereby validating our hypothesis. As the first step in this direction, we propose Local Amortized SAE (LocA-SAE), a method that groups polysemantically close latents based on the angular variance. This method is designed to balance the computational cost of per-sample optimization with the limitations of amortized inference. Our work provides insights for understanding SAEs and advocates for a paradigm shift in future research on polysemy disentanglement. The code is available https://github.com/wenjie1835/Local_Amotized_SAEs.

## 1 INTRODUCTION

Mechanistic Interpretability (MI) has emerged as a critical subfield in artificial intelligence, aiming to open the 'black box' through reverse-engineering the internal computational processes of neural networks to understand how models process information and make decisions (Bereska & Gavves, 2024). Unlike traditional black-box analysis, MI focuses on the specific mechanisms inside models, such as the roles of attention heads or activation patterns. However, the polysemy phenomenon presents challenges in understanding the mechanisms of components (Saphra & Wiegreffe, 2024). This manifests itself as a single neuron activating for multiple unrelated concepts, making it hard to attribute the neurons precisely (Bricken et al., 2023).

To tackle this challenge, researchers have introduced Sparse Autoencoders (SAEs), a tool to decompose activation vectors, aiming to extract a complete set of fundamental units that faithfully represent independent concepts, known as monosemantic features (Cunningham et al., 2023; Templeton et al., 2024). As a neural network implementation of sparse dictionary learning, unlike traditional sparse coding, which solves a regularized iterative optimization problem instance-wise (Fel et al., 2025; Tibshirani, 2013), SAEs employ an encoder to parameterize an inference network. This network learns a deterministic mapping function from input data to sparse codes through global training, thereby amortizing the optimization cost across all samples, a process known as amortized infer-

---

*Correspondence to Lijie Hu (Email Address: lijie.hu@mbzuai.ac.ae)

ence (O'Neill et al., 2024). This approach pursues a global optimum by end-to-end minimizing reconstruction error and sparsity penalty (Kissane et al., 2024).

Although efficient, amortized inference often sacrifices instance-level optimality for global constraints (O'Neill et al., 2024; Costa et al., 2025). In contrast, the monosemantic features desired by the MI community emphasize "instance-level optimality," where each sparse code robustly and accurately reflects a single concept (Bricken et al., 2023), which is similar to the instance-wise optimal solutions obtained through sample-by-sample iterative optimization in classical sparse coding like matching pursuit. The performance gap between the sparse codes derived from amortized inference-based SAEs and the ideal sparse codes expected for each sample is referred to as the amortization gap (O'Neill et al., 2024; Song et al., 2025). This gap represents a key price of amortized inference, as it introduces systematic suboptimality that conflicts with the instance-wise precision required for monosemantic features.

To understand the price of amortized inference, we first examine how the amortization gap manifests in various pathological phenomena observed in SAEs. These issues include **feature absorption**, whereby the shared encoder represents multiple concepts with a single latent to meet a global sparsity budget, and **feature splitting**, where a complex concept is approximated by multiple redundant latents to minimize average reconstruction error (Chanin et al., 2024). Another related issue is the **prevalence of dense latents**, which indicates overfitting to high-frequency activation patterns at the expense of per-sample sparsity (Bussmann et al., 2024a). The last phenomenon is **feature inconsistency**. It is the representation of a single concept by different latents under minor distribution shifts, which stems from the encoder's failure to find a stable optimum, instead converging to divergent local minima (Song et al., 2025; Leask et al., 2025). Collectively, these pathologies highlight a core trade-off: amortized inference achieves global efficiency by sacrificing the per-sample fidelity essential for faithful mechanistic interpretation.

Given the problems introduced by amortized inference in the task of disentangling polysemantic features, this paper argues against over-investment in fully-amortized paradigms, such as improvements to activation functions and gating mechanisms. Accordingly, we propose a locally amortized SAE variant, aimed at finding a trade-off between the computational cost of per-sample optimization and amortized inference. Our contributions can be summarized as follows.

1. We argue that the inherent misalignment between the global optimality emphasized by amortized inference-based SAEs and the instance-level optimality required for monosemantic features indirectly leads to numerous pathological phenomena in SAEs. From the perspective of training dynamics, we reveal that pursuing the global reconstruction-sparsity Pareto frontier does not yield improvements in monosemanticity. This issue stems from an architectural trade-off induced by the parameter sharing of amortized inference under global sparsity/reconstruction constraints. Specifically, this trade-off means that increasing the sparsity penalty not only fails to improve monosemanticity but instead exacerbates pathological phenomena such as dead latents, feature splitting, and feature absorption.
2. To verify the relationship between amortized inference and these pathological phenomena, we introduce semi-amortized and non-amortized methods as intervention and ablation experiments, respectively, to reduce reliance on amortized inference. Results show that these alternative approaches significantly reduce reconstruction errors and alleviate the dead latent problem. Furthermore, the extracted features exhibit superior performance in targeted concept removal tasks and enhanced controllability in model intervention tasks.
3. As the first step in this direction, we introduce Local Amortized Sparse Autoencoders (LocA-SAE). This method groups latent variables based on their angular variance and assigns an independent encoder and sparsity penalty to each group. This design aims to mitigate the unreasonable trade-offs caused by varying degrees of latent polysemanticity, while balancing the computational cost of instance-wise optimization against the limitations of amortized inference. Results indicate that LocA-SAE alleviates multiple pathological phenomena and even eradicates dead latents entirely, with only a marginal sacrifice in reconstruction performance.

## 2 RELATED WORK

**Sparse Autoencoders.** SAEs (Bricken et al., 2023), as a neural network implementation of dictionary learning, aim to learn sparse representations of input data, particularly useful in MI for

decomposing polysemantic activations in Large language models (LLMs) Yao et al. (2025); Yang et al. (2024). The core principle involves reconstructing input activations $\mathbf{x} \in \mathbb{R}^d$ using a learned overcomplete dictionary $\mathbf{D} \in \mathbb{R}^{d \times m}$ and sparse latent codes $\boldsymbol{\alpha} \in \mathbb{R}^m$, where the encoder maps $\mathbf{x}$ to $\boldsymbol{z}$ and the decoder reconstructs $\hat{\mathbf{x}} = \mathbf{D}\boldsymbol{z}$, minimizing reconstruction loss plus sparsity penalties to extract monosemantic features that resolve superposition (Elhage et al., 2022). SAEs' evolution starts with vanilla SAEs, employing $L_1$ penalties but facing dead features and shrinkage bias (Cunningham et al., 2023). To mitigate these, TopK SAEs enforce hard sparsity via top-$k$ pre-activations, reducing dead latents and enhancing scaling in models like GPT-4 (Gao et al., 2024). Building on this, BatchTopK SAEs relax constraints batch-wise, alleviating sparsity variance and boosting stability in high dimensions (Bussmann et al., 2024b). For shrinkage, Gated SAEs decouple detection and magnitude estimation via dual paths, yielding gains across hyperparameters in 7B models (Rajamanoharan et al., 2024a). Further refining $L_0$ sparsity, JumpReLU SAEs use discontinuous activations and straight-through estimators, achieving top fidelity on Gemma 2 9B (Rajamanoharan et al., 2024b). Addressing feature splitting, Matryoshka SAEs train nested increasing-width SAEs for hierarchical features and multi-resolution analysis (Bussmann et al., 2024a). Additionally, AdaptiveK SAEs dynamically tune $k$ for uneven distributions; P-anneal SAEs anneal penalties progressively to avoid early dead features (Yao & Du, 2025). Recent RouteSAEs extend to multi-layers with routers integrating residual activations, capturing cross-layer features for better interpretability (Shi et al., 2025). Additional related work can be found in the Appendix. B.

## 3 Preliminaries

### 3.1 Sparse Coding and Amortization-Based Sparse Autoencoders

Sparse Coding, also known as sparse dictionary learning (Olshausen & Field, 1996), aims to represent an input signal as a linear combination of a set of overcomplete basis vectors, while constraining the representation coefficients to be as sparse as possible. Formally, given an input vector $\mathbf{x} \in \mathbb{R}^d$ and an overcomplete dictionary matrix $\mathbf{D} \in \mathbb{R}^{d \times k}$ (where $k > d$), sparse coding seeks the optimal sparse code $\mathbf{z}^* \in \mathbb{R}^k$ that satisfies:

$$\mathbf{z}^* = \arg\min_{\mathbf{z}} \|\mathbf{x} - \mathbf{D}\mathbf{z}\|_2^2 + \lambda \|\mathbf{z}\|_0, \tag{1}$$

where $\|\cdot\|_0$ denotes the $\ell_0$ pseudo-norm, and $\lambda$ controls the sparsity strength. Since the $\ell_0$ optimization is NP-hard, the $\ell_1$ norm is often employed as a convex surrogate for the sparsity constraint in practice. Inference in this classical formulation is an iterative process performed on each sample, ensuring instance-specific optimality (Chen et al., 2001; Mallat & Zhang, 1993).

SAE emerges as a neural network implementation of this dictionary learning framework, designed to overcome the computational bottleneck of per-sample optimization. The core architecture of an SAEs consists of an encoder $f_\phi$ and a decoder. The encoder $f_\phi$ is typically a linear transformation followed by a non-linear activation function, which maps the input $\mathbf{x}$ to a sparse latent code $\mathbf{z} = f_\phi(\mathbf{x})$. The decoder then reconstructs the input using a learned dictionary matrix $\mathbf{D} \in \mathbb{R}^{d \times k}$: $\hat{\mathbf{x}} = \mathbf{D}\mathbf{z}$ (Braun et al., 2024). The training objective minimizes the reconstruction error plus a sparsity penalty:

$$\mathcal{L} = \|\mathbf{x} - \hat{\mathbf{x}}\|_2^2 + \lambda \|\mathbf{z}\|_1. \tag{2}$$

Different from the traditional sparse coding schemes to solve an optimization problem for each instance, Crucially, SAEs employ an amortized inference approach. A shared encoder network $f_\phi$ learns to approximate the posterior over the entire dataset. This approach amortizes the computational cost of inference across the training process. Consequently, SAEs replace the iterative, sample-specific optimization of classical sparse coding with efficient, global inference via an end-to-end feedforward network, enabling scalable application to large-scale data.

### 3.2 Pathological Phenomena in SAEs

The pursuit of monosemantic features through the amortized inference of SAEs often lead to several common pathological phenomena. These phenomena represent failures in the desired behavior of the learned dictionary and its latents. We briefly define these pathologies and the metrics used to quantify them, which are crucial for interpreting our subsequent analysis.

**Dead Latents.** A significant portion of latent units may rarely or never activate, indicating a failure to utilize the full capacity of the overcomplete dictionary and reducing the effective model size (Gao et al., 2024).

**Dense Latents.** Contrary to Dead Latents, some latents activate excessively frequently across inputs. These latents often correspond to common, non-specific directions or polysemantic combinations, violating the goal of sparsity and monosemanticity (Sun et al., 2025).

**Feature Splitting.** A single coherent concept may be represented by the activation of multiple similar or redundant features across different contexts. This fragmentation obscures the intended one-to-one mapping between features and concepts, reducing interpretability (Chanin et al., 2024).

**Feature Absorption.** A rarer concept can be "absorbed" into a more frequent one, whereby the encoder opts to represent both using the same dominant latent. This results in the suppression of the rarer concept's unique latent and compromises feature completeness (Chanin et al., 2024).

## 4 THE MISALIGNMENT OF AMORTIZED SAES FOR POLYSEMY DISENTANGLEMENT

In this section, we argue from two perspectives about the misalignment of amortized inference and the polysemy problem. The first is a mismatch in evaluation metrics: the global reconstruction-sparsity trade-off emphasized by amortized inference overlooks the instance-level trade-off highlighted by monosemanticity and may obscure certain pathological phenomena. The second is that the optimization approach of amortized inference tends to preserve latent variables with high marginal contribution (activation frequency) under global reconstruction-sparsity constraints, which conflicts with the instance-level semantic purity emphasized by monosemanticity.

### 4.1 AMORTIZATION GAP FROM THE PERSPECTIVE OF PARETO FRONTIER

The concept of the Amortization Gap is well-established in variational inference literature, referring to the systematic discrepancy between the approximate posterior learned by an amortized inference network and the true posterior (Kim et al., 2018; Marino et al., 2018). In SAEs, O'Neill et al. (2024) define this gap as the systematic discrepancy between the latent representations predicted by a shared encoder $f_\phi(\mathbf{x})$ under the amortized inference framework and those obtained by instance-wise optimization.

Mathematically, for a given input $\mathbf{x}$, let $\mathbf{z}_a = f_\phi(\mathbf{x})$ be the amortized sparse code produced by the SAE encoder, and $\mathbf{z}_o$ be the optimal sparse code obtained via per-sample optimization:

$$\mathbf{z}_o = \arg\min_{\mathbf{z}} \|\mathbf{x} - \mathbf{Dz}\|_2^2 + \lambda\|\mathbf{z}\|_1. \tag{3}$$

The amortization gap can be formalized as the difference in the objective function values:

$$\Delta(\mathbf{x}) = \left(\|\mathbf{x} - \mathbf{Dz}_a\|_2^2 + \lambda\|\mathbf{z}_a\|_1\right) - \left(\|\mathbf{x} - \mathbf{Dz}_o\|_2^2 + \lambda\|\mathbf{z}_o\|_1\right). \tag{4}$$

This gap $\Delta(\mathbf{x}) \geq 0$ quantifies the suboptimality per sample, with the average gap over a dataset providing a global metric: $\bar{\Delta} = \frac{1}{N}\sum_{i=1}^{N}\Delta(\mathbf{x}_i)$.

$\bar{\Delta}$ reflects a fundamental trade-off between efficiency and precision, O'Neill et al. (2024) proves that there is a theoretical minimum from the compressed sensing theory, and attributes it to their linear-nonlinear structure. Here, we describe the $\bar{\Delta}$ as the "distance" between the *reconstruction/sparsity Pareto frontier* of SAEs (their optimal trade-off curve between reconstruction error and sparsity) and the optimal frontier defined by sparse codes from unconstrained sparse inference algorithms (e.g., sparse coding, which solves Eq. 3 per sample). Therefore, the $\bar{\Delta}$ can serve as a metric to quantify the global Pareto improvement in reconstruction/sparsity. This is exactly the goal of improvements in most current SAE variants: to minimize reconstruction error while simultaneously maximizing sparsity.

However, minimizing $\bar{\Delta}$ to pursue the global reconstruction-sparsity Pareto frontier may incur an overlooked cost: it evaluates the trade-off at the level of the entire dataset, whereas monosemanticity emphasizes this trade-off at the instance level. Consequently, the convergence of $\bar{\Delta}$ does not necessarily imply an improvement in monosemanticity and may instead mask the inherent trade-offs among pathological phenomena, thereby misleading researchers.

## 4.2 THE PARADOX BETWEEN GLOBAL OPTIMALITY AND MONOSEMY

Amortization-based encoder are trained to minimize an expected reconstruction–sparsity tradeoff, the shared encoder is encouraged to learn "high-frequency, cross-domain reusable" directions to reduce overall error (Kim et al., 2018; Cremer et al., 2018). However, the evaluation standard for monosemantic features emphasizes per-example semantic purity, robustness, and stitchability (Karvonen et al., 2025). This inherent tension indirectly leads to many pathological phenomena in SAEs. For instance, when the data distribution is long-tailed or multimodal, minimizing expected error and preserving per-example semantic atomicity often cannot be achieved simultaneously. In such cases, the optimizer compromises by sacrificing semantic consistency and trigger completeness for some samples to favor reconstruction accuracy. This mechanism underlies phenomena such as feature splitting and feature absorption, which not only waste dictionary capacity but also reduce interpretability (Chanin et al., 2024).

Furthermore, if the goal is to discover "canonical units," current SAEs still fail to converge to a unique and stitchable set of features. Through stitching and meta-SAE experiments, Leask et al. (2025) demonstrated that latent variables learned by different SAEs on the same dataset do not form a unified atomic set, reinforcing the conclusion that "global mean optimality $\neq$ per-sample and cross-setup optimality." Mechanistically, neural networks encode unrelated semantics in nearly orthogonal directions to accommodate more sparse features within limited dimensions, resulting in neuron-level polysemy and "space sharing." This makes the exact way in which rare concepts are absorbed or split highly sensitive to initialization and mild distribution shifts, so the resulting features are difficult to stitch across runs into a run-invariant basis. In contrast, the instance-wise optimum $z_o(x)$ depends only on the dictionary and sparsity penalty and therefore defines a more stable target. Based on such premises, a monosemantic objective that stresses instance-wise atomicity is intrinsically in tension with a global objective that optimizes average reconstruction (Elhage et al., 2022).

We therefore hypothesize that this fundamental misalignment does not merely introduce a performance gap but systematically distorts the feature learning process, forcing the model into a regime of unreasonable trade-offs among several pathological phenomena. Specifically, when the data distribution is long-tailed or multimodal, minimizing expected error and preserving per-example semantic atomicity often cannot be achieved simultaneously. In such cases, the optimizer compromises by sacrificing semantic consistency and trigger completeness for some samples to favor reconstruction accuracy. This mechanism underlies phenomena such as feature splitting and feature absorption, which not only waste dictionary capacity but also reduce interpretability (Chanin et al., 2024). Dead latents may emerge as the global constraint prunes low-frequency directions, while dense latents persist due to overfitting to common patterns, further exacerbating the misalignment.

## 4.3 EVIDENCE FROM TRAINING DYNAMICS

To empirically validate our hypothesis that these pathologies stem from this paradox, we examine the training dynamics of SAEs. By tracking the evolution of key metrics under varying sparsity constraints, we aim to understand the trade-offs among these phenomena, demonstrating how the global amortized objective fails to enhance monosemanticity.

**Experimental Setup.** We use SAEBench's open-source implementation, which includes both Standard SAE and Top-k SAE models trained on the $resid\_post$ of the 12th layer of Gemma-2-2B over the *Pile-uncopyrighted* dataset (Karvonen et al., 2025). Each variant incorporates six different sparsity strengths and checkpoints from seven distinct training steps. The evaluation metrics for these pathological phenomena are detailed in Section 3.2. Notably, for the calculation of the amortization gap, the optimal sparse code $z_o$is obtained via 200 iterations of the ISTA (Iterative Shrinkage-Thresholding Algorithm).
**Evaluation Matrices.** We employ a suite of metrics including the *Dead Rate*, *Dense Rate*, *Absorption Rate*, and *ΔF1* (for feature splitting), alongside standard measures like *Normalized Mean Squared Error (NMSE)* and the *Amortization Gap ($\bar{\Delta}$)*. The formal definitions and mathematical details of all evaluation metrics are provided in Appendix I (Table. 9).

**Observations.** We compare the dynamic behaviors of various pathological phenomena in two variants of SAEs under different sparsity levels: Standard SAE and Top-k SAE, as illustrated in Figures 1 and 2, detailed experimental results can be seen in the Table. 4 and 5 in the Appendix F. Our

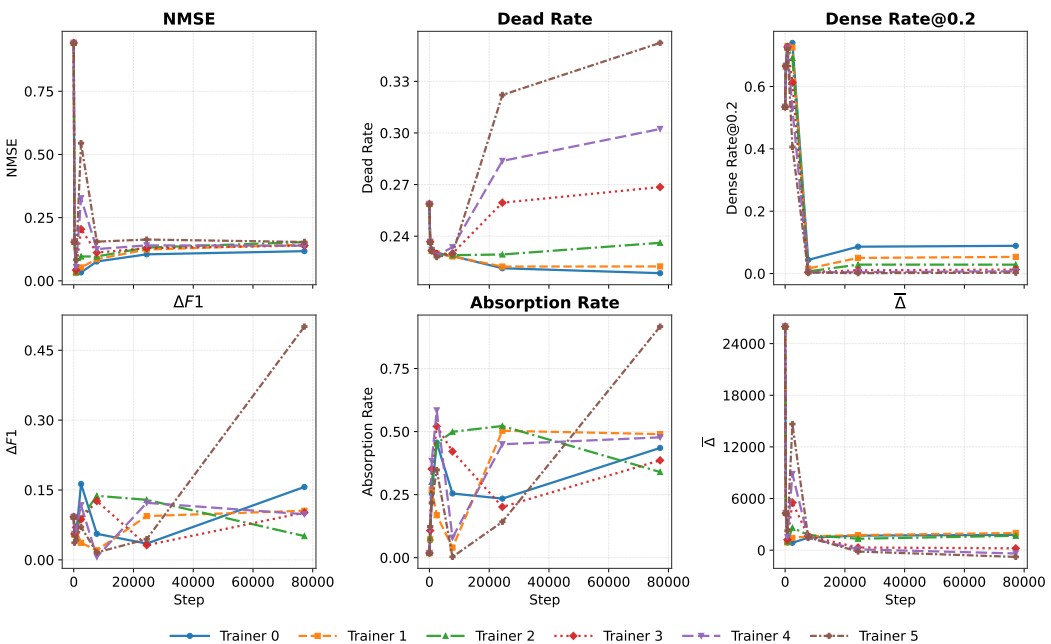

Figure 1: Variation of Different Pathological Phenomena Corresponding to Standard SAE at Different Sparsity Levels from the Perspective of Training Dynamics. Sparsity gradually increases from $Trainer$ 0 to $Trainer$ 5.

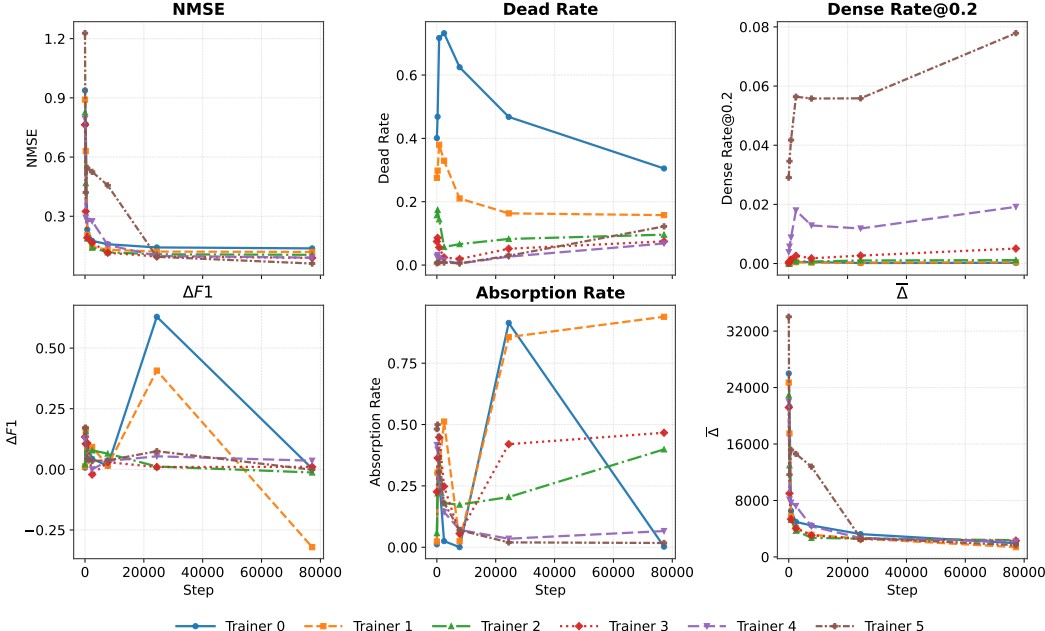

Figure 2: Variation of Different Pathological Phenomena Corresponding to Top-k SAE at Different Sparsity Levels from the Perspective of Training Dynamics. Sparsity gradually increases from $Trainer$ 5 to $Trainer$ 0.

analysis reveals that the training process forces an unreasonable balancing of pathologies, driven by the conflict between a shared encoder and a global sparsity budget.

**Sparsity exacerbates dead latent without resolving dense ones.** As shown in the subfigure of NMSE and Dead Rate of Figure 1 and 2, increasing the sparsity penalty leads to a steady rise in

the NMSE and Dead Rate. This occurs because the global objective prioritizes latents with high marginal gain (activation frequency). Low-frequency latents are pruned first as sparsity costs increase. However, the Dense Latent Rate was only slightly mitigated. Additionally, as the number of training steps increased, the Dead Rate and Dense Rate did not show a complementary trend. Taking Trainers 3-5 of the Top-k SAE as an example (Figures 2_Dead Rate, Dense Rate@0.2), after the initial training phase, these two metrics even demonstrated a tendency to be exacerbated simultaneously. indicating that the common, high-utility directions are preserved due to the strong reconstruction constraint. This creates a difficult trade-off: increased sparsity fails to clean up overly dense latents while simultaneously killing off more niche features.

**Feature splitting and absorption emerge as compensatory mechanisms.** The global reconstruction-sparsity objective often finds it advantageous to represent a concept with multiple splintered features (increasing $\Delta$F1 in Fig. 1) or to assign the variance of a rare feature to a more frequent, absorbing latent (Fig. 1_Absorption Rate). For example, in the mid-to-late stages of training for the Standard SAE under high sparsity, we observe a significant spike in both Absorption Rate and $\Delta$F1. This suggests the model is compensating for the tight sparsity budget by making representations less interpretable, not more monosemantic.

**The opposite trend between monosemanticity and amortization gap.** Figure 1 shows that $\bar{\Delta}$ decreases consistently during training for all sparsity levels, indicating that the amortized mapping learned by the encoder does lead to better global Pareto improvements. However, this trend does not correlate with improvements in monosemanticity metrics (Dead, Dense, Absorption, $\Delta$F1). This demonstrates that optimizing the global amortized objective is not sufficient for learning instance-optimal, monosemantic features, and these pathological phenomena are inherent to the paradigm. It also points out that the global reconstruction-sparsity Pareto frontier pursued by most current SAEs variants is directionally biased.

**Top-K SAEs alleviate but do not eliminate the trade-off.** The Top-K SAE (Fig. 2), with its hard gating mechanism, shows a drastically reduced Dense Rate and avoids the worst of the dead feature problem. However, it still exhibits a trend of increasing with increasing sparsity in Absorption and $\Delta$F1 during training (Fig. 2_$\Delta$F1, Absorption Rate). It suggests that while modifications to activation functions and gating mechanisms may alleviate specific pathological phenomena, the root conflict stemming from amortization remains unresolved.

In conclusion, the pathological phenomena are inextricably linked, they are not independent failures but interrelated symptoms of a shared encoder competing for a limited global budget. Pursuing a better global Pareto frontier on reconstruction and sparsity within the amortized paradigm comes at the direct expense of monosemanticity. These findings support our hypothesis that the pathologies originate from the fundamental paradox between global and instance-level optimality, providing empirical evidence for the inherent limitations of amortized inference in SAEs.

## 5 DO PATHOLOGICAL PHENOMENA REALLY STEM FROM AMORTIZED INFERENCE?

To further verify this attribution, we conducted intervention and ablation experiments. Specifically, we explores semi-amortized and non-amortized approaches by reducing or eliminating reliance on shared encoders, aiming to demonstrate whether instance-level optimization can mitigate the observed pathological phenomena. Finally, we propose an intermediate approach to balance computational cost and amortized inference named LocA-SAE: a variant of SAE that groups latent variables based on angular variance and assigns each group an independent encoder for locally amortized encoding at varying sparsity levels.

### 5.1 EXPERIMENTAL SETUP

For this purpose, we reproduce part of the work from SAEBench (Karvonen et al., 2025). Specifically, we train four different SAE architectures, Standard SAE, JumpReLU SAE, Gated SAE, Top-k SAE, BatchTopK SAE and Matryoshka SAE, which use the $resid\_post$ of the 8th layer of the Pythia-160m-deduped model processed on the monology/pile-uncopyrighted dataset, while for Gemma-2-2b, the 12th layer is used. The same $resid\_post$ is also used to train LoC-SAE. Detailed config can

be found in the Appendix H, and the evaluation metrics remain consistent with 3.2. The test data is the first 10,000 tokens from the training data loaded via streaming.

## 5.2 METHODOLOGY

Standard SAEs force a single encoder $W_{\text{enc}}$ to approximate instance-optimal codes for all latents simultaneously, inducing pathological trade-offs. We introduce LocA-SAE, which retains a single shared dictionary but replaces the global encoder with a small set of group-wise encoders at varying sparsity levels.

**Angular variance based Latent Grouping.** We group latent units based on angular variance. This metric is defined as $\text{AVar}_j = 1 - \|\mu_j\|_2$, which quantifies the diversity of activating samples and thereby indirectly represents the polysemanticity intensity of the latent unit, where $\mu_j$ denotes the mean direction of all normalized inputs that activate latent unit $j$. We then sort the latent units by $\text{AVar}_j$ and partition the index set $\{1, \ldots, m\}$ into $G = 8$ contiguous groups $\mathcal{G}_1, \ldots, \mathcal{G}_G$.

**Architecture and Training.** The architecture comprises a global decoder $D \in \mathbb{R}^{d \times m}$ and discrete group encoders $W_{\text{enc}}^{(g)}$. Each $W_{\text{enc}}^{(g)}$ computes pre-activations $u^{(g)}$ from input $x$, which is then sparsified into a group code $z^{(g)}$ via a group-specific Top-$k_g$ operation and ReLU. The budgets are configured as $(k_1, \ldots, k_8) = (6, 5, 4, 3, 3, 2, 2, 1)$, aimed at enforcing higher sparsity on low-variance groups. Global reconstruction is given by $\hat{x} = Dz$. This design preserves amortized inference efficiency while enabling heterogeneous sparsity. Training is a four-stage process: pretraining a global SAE; grouping latents by angular variance; initializing group encoders via weight copying; and fine-tuning all parameters under group-specific $k_g$ constraints.

## 5.3 INTERVENTION AND ABLATION

To separate the effects of amortization from architectural choices, we treat the inference method as an intervention variable. We compare LocA-SAE against Fully-Amortized baselines and two "intervention" methods defined as follows:

**Semi-Amortized**: A balanced hybrid method that begins with the quick prediction from the fully-amortized encoder but then fine-tunes it with a few steps of sample-specific optimization.
**Non-Amortized**: A complete per-sample optimization starting from scratch, without relying on the shared encoder at all.

**Implementation Details:** Given a token activation $x \in \mathbb{R}^d$ and dictionary $\mathbf{D} \in \mathbb{R}^{d \times m}$, we solve the nonnegative sparse coding problem:

$$L_\lambda(x, z) = \frac{1}{2}\|x - \mathbf{D}z\|_2^2 + \lambda\|z\|_1, \quad z \geq 0. \tag{5}$$

The amortized encoder computes an initial code $z^{(0)} = \max(\mathbf{W}^\top x + b, 0)$, with parameters $\mathbf{W}$ and $b$. For Top-$K$, we select the top-$K$ pre-activations before ReLU.

Semi-amortized inference refines $z^{(0)}$ over $T_{\text{semi}}$ ISTA steps:

$$z^{(t+1)} = \max\left(z^{(t)} - \alpha\big(\mathbf{D}^\top(\mathbf{D}z^{(t)} - x) + \lambda\mathbf{1}\big), 0\right), \tag{6}$$

with step size $\alpha \approx 1/\|\mathbf{D}\|_2^2$ from power iteration.

Non-amortized inference runs $T_{\text{ista}}$ ISTA steps from zero:

$$z^{(t+1)} = \max\left(z^{(t)} - \alpha\,\mathbf{D}^\top(\mathbf{D}z^{(t)} - x) - \alpha\lambda\mathbf{1}, 0\right). \tag{7}$$

For fair sparsity pattern comparison, we calibrate $\lambda$ per method on a held-out set to match the amortized baseline's activation density (e.g., Dense@0.1).

## 5.4 EMPIRICAL ANALYSIS

Table 6 in Appendix F.2 presents the pathological phenomenon metrics for various SAE variants under different amortization patterns, evaluated on the Pythia-160m-deduped (layer 8) and Gemma-2-2b (layer 12) models. Figures 3 and 4 illustrate the variations in key metrics across these patterns

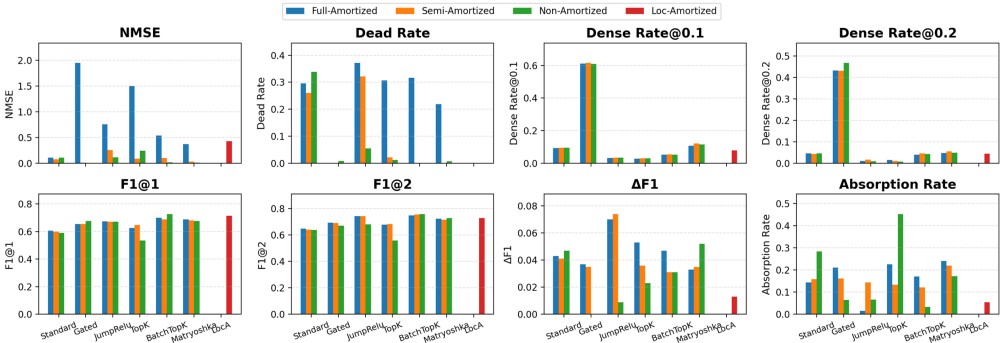

Figure 3: Pathological Phenomenon Indicators Corresponding to SAE Variants Under Different Amortization Models (Pythia-160m-deduped/ Layer8)

and variants. Overall, we observe consistent trends: reducing reliance on full amortization via semi-amortized interventions, non-amortized ablations, or the proposed Loc-Amortized approaches generally alleviates multiple pathologies. Specifically, non-amortized ablations and semi-amortized interventions achieve lower NMSE and more stable sparsity control in the majority of cases. In contrary, full-amortized often exhibits higher reconstruction errors and poorer sparsity, as the shared encoder prioritizes global efficiency over instance-level optimality. This gap is evident in variants like JumpReLU and GatedSAE, where non-amortized ablations reduce NMSE by up to 80%. Semi-amortized interventions mitigate this partially by refining amortized initializations with per-sample optimization, while non-amortized ablations ensure fully independent solutions, albeit at increased computational cost. The Loc-Amortized pattern in LocA-SAE strikes a balance, alleviating numerous pathological phenomena of full-amortized without requiring per-sample optimization, demonstrating competitive $\Delta F1$ and Absorption Rate, even eliminating dead latent completely. These results validate amortization as a root cause of pathologies, demonstrating that hybrid or localized approaches can resolve trade-offs like reconstruction-sparsity conflicts observed in training dynamics. We additionally perform an ablation over the number of refinement steps in the semi-amortized BatchTopK SAE (Appendix D). We find that increasing the number of steps yields monotonic but quickly saturating gains in NMSE, while the pathological metrics remain essentially unchanged, indicating that our conclusions are robust to the specific refinement budget and that a moderate number of steps already offers a good accuracy–compute trade-off.

Among the seven SAE architectures, GatedSAE consistently exhibits high dense rates and near-zero dead rates, indicating efficient feature utilization but at the expense of sparsity, which may compromise monosemanticity. JumpReLU and TopK, conversely, enforce stronger sparsity but suffer from elevated NMSE and absorption in full-amortized settings. BatchTopK and Matryoshka show promising behaviors: BatchTopK achieves low NMSE in semi-amortized interventions and non-amortized ablations with zero dead latents, suggesting improved stability through batch-wise sparsity relaxation. Matryoshka, designed for hierarchical features, NMSE decreases significantly as reliance on full-amortized on diminishes. and also exhibited lower Absorption Rate and $\Delta F1$. LocA-SAE, our proposed Loc-Amortized variant, outperforms many baselines by eliminating dead latents entirely while maintaining low $\Delta F1$ and absorption, demonstrating that group-wise encoders with angular variance based partitioning enable heterogeneous sparsity without sacrificing monosemanticity. As a novel methodological framework, LocA-SAE bridges the efficiency of global amortization with the precision of instance-level optimization by grouping inputs into localized encoders, which is particularly effective in long tailed data distributions as it allows rare patterns to be captured independently in dedicated sub-encoders, thereby enhancing feature monosemanticity. Compared to the full-amortized baselines, LocA-SAE completely eliminates dead latents and substantially reduces absorption and splitting, at the cost of a modest increase in NMSE. Relative to the non-amortized ISTA baseline, LocA-SAE achieves similar improvements in pathological metrics while avoiding any per-sample iterative optimization. This places LocA-SAE on a favorable point of the fidelity–compute trade-off curve. These architectural comparisons reinforce that while tweaks like gating or hierarchy mitigate specific issues under full amortization, broader reductions in parameter sharing via semi-amortized interventions, non-amortized ablations, or local-amortization yield consistent improvements across variants, highlighting shared encoding as the primary bottleneck.

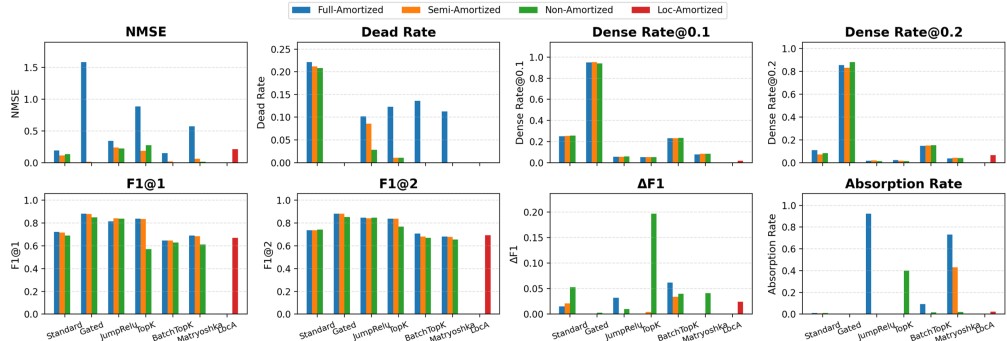

Figure 4: Pathological Phenomenon Indicators Corresponding to SAE Variants Under Different Amortization Models (Gemma-2-2b/ Layer12)

Additionally, several outliers deserve attention. For instance, *Gemma-2-2b/L12, JumpReLU, Full-Amortization* shows an abnormally high *Absorption*; switching to *Semi/Non* immediately drives it to 0, indicating primary-detector under-firing induced by single-step amortization. A few per-example optimization steps correct this mismatch (Fig. 4_Absorption Rate). Furthermore, GatedSAE under full amortization shows very high NMSE (>1.5), suggesting issues like over-regularization or encoder underfitting, which are mitigated in the semi- and non-amortized patterns. These outliers serve as direct evidence of amortization's problems, highlighting the advantages of semi- and non-amortization in instance-level optimization and further confirming that amortization leads to pathologies that can be alleviated by reducing parameter sharing. However, in a few cases, the reduced reliance on full amortization has also been accompanied by some negative results. This might be attributed to the inherent conflict between TopK hard constraints and ISTA soft-thresholding, combined with the loss of cross-sample semantic consistency resulting from the lack of global sharing, which subsequently triggers the absorption or splitting of rare features.

## 6 CONCLUSION AND DISCUSSION

Numerous recent variants of SAEs have emerged to mitigate pathological phenomena in polysemantic feature disentanglement, primarily targeting the reconstruction-sparsity trade-off. However, most of them have overlooked the trade-off between monosemanticity and this reconstruction-sparsity balance. In this study, we demonstrate that within amortization-based encoding frameworks, improvements along the reconstruction-sparsity Pareto Frontier do not lead to better monosemanticity. On the contrary, it comes at the expense of dictionary capacity and monosemanticity, while also inducing several pathological phenomenon. Furthermore, our intervention and ablation studies demonstrate that reducing reliance on full amortization not only consistently improves reconstruction performance and mitigates the dead latent problem but also yields features that are superior in targeted concept removal tasks and offer enhanced controllability in model interventions. However, since per-sample optimization exacerbates the scalability issues of SAEs, we propose LocA-SAE, a method that employs local amortization by grouping latents based on their angular variance, thereby balancing the computational cost of instance-wise optimization against the limitations of full amortization. Experimental results show that LocA-SAE not only completely eliminates dead latents but also significantly alleviates feature splitting and feature absorption. Based on this, we contend that various pathologies observed in SAEs stem not only from their unsupervised learning paradigm but also from the fully amortized inference inherent in their architecture. While modifications to gating mechanisms or activation functions may mitigate specific issues, they fail to resolve the fundamental conflict between parameter-sharing encoding and the instance-level optimality required for monosemantic features.

## ACKNOWLEDGMENT

Lijie Hu and Wenjie Sun is supported by the funding BF0100 from Mohamed bin Zayed University of Artificial Intelligence (MBZUAI). Di Wang if supported in part by the funding BAS/1/1689-01-

01,RGC/3/7125-01-01, FCC/1/5940-20-05, FCC/1/5940-06-02, and King Abdullah University of Science and Technology (KAUST) – Center of Excellence for Generative AI, under award number 5940 and a gift from Google.

## ETHICS STATEMENT

This work explores theoretical and empirical limitations of amortized inference in sparse autoencoders for mechanistic interpretability, advocating for alternative encoding paradigms. We anticipate no direct negative societal impacts from this research, as it focuses on improving the transparency and controllability of AI models. Enhanced monosemantic features could contribute to safer AI systems by facilitating better detection of biases, errors, or unintended behaviors in large language models. However, we acknowledge that advancements in interpretability tools might be dual-use; for instance, they could potentially aid in reverse-engineering models for malicious purposes, such as crafting adversarial attacks. To mitigate this, we emphasize ethical deployment and encourage open discussions on responsible AI research. All experiments use publicly available, uncopyrighted datasets, and no human subjects or sensitive data were involved.

## REPRODUCIBILITY STATEMENT

To ensure reproducibility, we provide detailed descriptions of our experimental setup in Sections 3 and 4, including hyperparameters, training procedures, and evaluation metrics. We utilize the open-source SAEBench framework for training Standard SAE and Top-k SAE variants on the 12th-layer residual post activations of Gemma-2-2B, using the Pile-uncopyrighted dataset. Checkpoints from seven training steps across six sparsity levels are analyzed. For semi-amortized and non-amortized methods, we employ ISTA with 200 iterations for optimal sparse codes. Code for experiments, including custom metrics (e.g., Dead Rate, Dense Rate, Absorption Rate, and $\Delta F1$), is based on SAEBench and will be released anonymously upon submission via a public repository. All results can be replicated with standard hardware (a single NVIDIA 5090D GPU for training).

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

# Appendices

## A    LLMS USAGE IN THE PAPER

LLMs were used only occasionally to help polish the writing (propose new words, grammar and spelling correction). All technical ideas, experimental designs, analyses, conclusions, writing were developed and carried out entirely by the authors. The authors have full responsibi1lity for the final text.

## B    RELATED WORK

### B.1    MECHANISTIC INTERPRETABILITY AND SUPERPOSITION HYPOTHESIS

MI aims to understand how neural networks process and store information by reverse-engineering their internal computational processes (Bereska & Gavves, 2024; Wang et al., 2025b;a; Yang et al., 2025a; Zhang et al., 2025b). The rise of this research field marks a shift from traditional behaviorist, black-box analysis methods toward an exploration of internal mechanisms akin to cognitive neuroscience (Sharkey et al., 2025). Unlike functional explanations that focus on the overall behavior of models, MI focuses on parsing the specific computational mechanisms inside neural networks, including the specific functions of components such as attention heads, feedforward neural networks, and activation patterns (Saphra & Wiegreffe, 2024; Yang et al., 2025b; Zhang et al., 2025a; Dong et al., 2025; Su et al., 2025; Zhang et al., 2024). However, the existence of polysemantic features poses a major challenge to component attribution, where a single neuron responds to multiple unrelated concepts simultaneously. Inspired by the linear representation hypothesis (Arora et al., 2016), the Superposition hypothesis proposes that polysemantic features are composed of linear combinations of multiple independent concepts (Elhage et al., 2022), thereby initiating research on disentangling polysemantic features based on SAEs (Bricken et al., 2023; Gao et al., 2024; Karvonen et al., 2025).

### B.2    AMORTIZED INFERENCE

Amortized Inference is a method that uses a learned parameterized function to approximate the posterior distribution of latent variables (Shu et al., 2018). Its core idea is to "amortize" the computational cost of inference across multiple data instances, thereby avoiding computationally expensive iterative optimization for each sample. This paradigm has become relatively mature in variational autoencoders (VAEs) (Margossian & Blei, 2023). However, this gain in efficiency comes at the cost of compromised representation quality, known as the amortization gap. Zhang et al. (2022a) demonstrated that amortized inference leads to a degradation in approximation quality, i.e., a systematic discrepancy between the learned posterior distribution and the true posterior. This gap is particularly pronounced in complex models and large-scale datasets, manifesting as reduced generalization ability of the learned encoder, especially when faced with out-of-distribution samples. Similarly, O'Neill et al. (2024) proved the inherent suboptimality of SAEs and demonstrated that this amortization gap stems from their linear-nonlinear encoder structure based on compressed sensing theory. Current methods to mitigate the amortization gap can be broadly categorized into five types: (1) Semi-amortization, which starts from the encoder's code and applies a few per-sample optimization steps without abandoning end-to-end training (Kim et al., 2018; Marino et al., 2018); (2) Encoder-side structure, which increases expressivity to better approximate the per-sample solution map; (3) Amortized sampling, which distills MCMC into fast inference networks to balance fidelity and cost (Li et al., 2017); (4) Regularization design, which reduces mismatch and $L_1$-shrinkage via loss/constraint choices (Shu et al., 2018; Burda et al., 2015); and (5) Local amortization, which shifts shared inference from the global level to sub-distribution levels to avoid global dependencies (Wu et al., 2020; Liu & Liu, 2020). However, these mitigations that directly adapted to SAEs for polysemy disentanglement are still limited. This stems from the fact that VAEs are primarily applied to image generation tasks, where the latent space emphasizes smoothness and continuity to ensure sampling quality and generalization capability, prioritizing distribution-level optimality (Zhang et al., 2022b). In contrast, the latent space of SAEs serves the purpose of interpretability, emphasizing sparsity, atomicity, and discreteness, prioritizing instance-level optimality. While amortized inference emphasizes global optimality, which aligns well with the distribution-level optimality needed for VAEs, but unsuitable for the instance-level optimality required by SAEs.

| SAE Variant | Encoding Pattern | $\Delta\mathrm{Acc_{target}}\uparrow$ | $\Delta\mathrm{Acc_{non\text{-}target}}\downarrow$ | Selectivity $\uparrow$ |
|---|---|---|---|---|
| Top-k | Fully-amortized | 0.303 | 0.058 | 0.245 |
| Top-k | Non-amortized | 0.531 | 0.047 | 0.484 |
| Top-k | Semi-amortized | **0.552** | **0.044** | **0.508** |
| Gated | Fully-amortized | 0.358 | 0.066 | 0.292 |
| Gated | Non-amortized | 0.404 | 0.051 | 0.353 |
| Gated | Semi-amortized | 0.422 | 0.048 | 0.374 |
| JumpReLU | Fully-amortized | 0.321 | 0.063 | 0.258 |
| JumpReLU | Non-amortized | 0.374 | 0.057 | 0.317 |
| JumpReLU | Semi-amortized | 0.392 | 0.050 | 0.342 |
| Standard | Fully-amortized | 0.283 | 0.071 | 0.212 |
| Standard | Non-amortized | 0.315 | 0.062 | 0.253 |
| Standard | Semi-amortized | 0.336 | 0.052 | 0.284 |

Table 1: Targeted Probe Perturbation (TPP) on Pythia-160M-deduped (layer 8, $m$=16,384, $M$=100).

## C PERFORMANCE OF DIFFERENT AMORTIZATION PATTERNS IN DOWNSTREAM TASKS

### C.1 TARGETED PROBE PERTURBATION (TPP)

To evaluate the directional controllability of features learned by SAEs under different amortization paradigms, we conduct the TPP experiment. We first train linear probes on a fixed model layer's activations to identify specific concepts. Subsequently, dictionary latents are ranked via an attribution score, and a select subset is ablated from the residual stream. The core evaluation measures the probe's performance degradation on the target class while verifying the stability of non-target classes Karvonen et al. (2024).

**Model and Data.** The experiment is based on the **Pythia-160M-deduped** model, using activations $x \in \mathbb{R}^{768}$ from the `resid_post` of layer 8. We evaluate four SAE variants (Standard, Gated, JumpReLU, and Top-k) with a dictionary size of $m = 16,384$. The TPP experiment utilizes the **AG News** dataset, for which we use the representation of the last non-padding token from each sample as the probe's input.

**Inference Paradigms, Attribution, and Intervention.** The experiment compares three inference paradigms: **fully-amortized**, **semi-amortized**, and **non-amortized**, all sharing a common decoder dictionary $D$. For targeted intervention, we first compute an attribution score $s_j$ for each latent $j$ with respect to a given class probe's weight vector $w \in \mathbb{R}^d$:

$$s_j = \langle D_{:,j}, w \rangle \cdot \left( \mathbb{E}[z_j \mid y=1] - \mathbb{E}[z_j \mid y=0] \right).$$

This score jointly considers the alignment of a dictionary atom with the probe's direction and the feature's class-conditional activation difference. We select the top-$M$ latents according to $|s_j|$ to form an index set $S$ and perform *zero-ablation* on the activation vector $x$:

$$x' = x - D_{:,S} z_S,$$

which subtracts the reconstructed components corresponding only to the latents in $S$.

**Metrics.** We employ the following metrics: (i) **On-target Drop** ($\Delta\mathrm{Acc_{target}}$): The accuracy decrease on the target class, $\mathrm{Acc_{base}} - \mathrm{Acc_{ablated}}$. (ii) **Off-target Leakage** ($\Delta\mathrm{Acc_{non-target}}$): The mean accuracy change across non-target classes, measuring intervention precision. (iii) **Top-$M$ Curve**: $\Delta\mathrm{Acc_{target}}$ as a function of the number of ablated latents $M$.

**Observation** As shown in Table. 1, the results of the Targeted Probe Perturbation experiment demonstrate that semi-amortized and non-amortized inference methods consistently outperform the fully-amortized approach across all tested Sparse Autoencoder variants. Specifically, for all four

SAE architectures—Top-k, Gated, JumpReLU, and Standard—adopting semi- or non-amortized inference leads to a significant increase in the target class accuracy drop and better control over off-target leakage, resulting in marked gains in the selectivity metric.

These results resonate with the core argument in Section. 4.3 of the main text: the global optimality pursued by fully-amortized inference comes at the cost of instance-level semantic purity. From the perspective of a downstream intervention task, this experiment confirms that reducing reliance on the parameter-shared encoder significantly mitigates the amortization gap and enhances the monosemanticity and intervenability of the learned features, thereby providing strong empirical support for the paper's advocacy of "reducing over-investment in amortization-based encoding methods."

## C.2 GENERATIVE INTERVENTION SCORING (GIS)

To further assess the interpretability of SAE latents, we perform generative interventions. We manipulate a targeted set of latents during autoregressive generation and evaluate the effect on the model's output using an external scoring language model (LM). The intervention strength is rigorously calibrated to ensure fair comparisons across different amortization paradigms.

**Model and Data.** We use the same setup as in the TPP experiment: **Pythia-160M-deduped** at `resid_post`, layer 8. Latent sets for intervention are selected from the high-attribution features identified via TPP. To prevent informational leakage, a separate, base LM is employed as a scorer. Prompts are designed to elicit interpretable phenomena, such as numeracy and pronoun resolution.

**Intervention and Calibration.** Interventions can be either *zero-ablation* or *additive*. To ensure comparability, the intervention strength, controlled by a multiplier $\alpha$, is calibrated for each setup. Specifically, we use a binary search to find an $\alpha$ such that the mean per-token KL divergence between the clean and intervened next-token distributions matches a predefined target $\kappa \in \{0.10, 0.33, 1.00\}$.

**Metrics.** The primary metric is the **Intervention Score** ($S$), which quantifies the change in log-probability of a target hypothesis $\phi$ as evaluated by the scorer model $p_M$:

$$S = \mathbb{E}\big[\log p_M(\phi \mid g_I) - \log p_M(\phi \mid g)\big],$$

where $g_I$ and $g$ denote the intervened and clean generations, respectively. We compare $S$ across all SAE variants and inference paradigms at matched KL divergence levels.

**Observations** The results (Table. 2) show that semi-amortized and non-amortized inference paradigms consistently achieve higher intervention scores ($S$) than full-amortized ones across all SAE variants (TopK, Gated, JumpReLU, Standard). For example, with TopK SAE at KL target $\kappa = 1.00$, semi-amortization yields 1.968, non-amortization 1.781, and full-amortization 1.412. This trend persists across variants and KL levels. Furthermore, semi-amortization slightly outperforms non-amortized in most cases, indicating that limited per-sample optimization improves feature quality while balancing efficiency.

These findings align with Section 4.3, where global optimality in fully-amortized inference sacrifices instance-level semantic purity, favoring high-frequency latents and reducing intervention precision. In contrast, semi- and non-amortized methods mitigate the amortization gap, enhancing monosemanticity and intervenability. In summary, the GIS experiment supports reducing reliance on amortized inference for better polysemy disentanglement.

| SAE | Encoding Pattern | $S$ @ $\kappa$=0.10 | $S$ @ $\kappa$=0.33 | $S$ @ $\kappa$=1.00 |
|---|---|---|---|---|
| TopK | Semi-amortized | 0.231 | 0.646 | 1.975 |
| TopK | Unamortized | 0.224 | 0.653 | 1.812 |
| TopK | Amortized | 0.168 | 0.472 | 1.415 |
| Gated | Semi-amortized | 0.214 | 0.599 | 1.792 |
| Gated | Unamortized | 0.206 | 0.552 | 1.796 |
| Gated | Amortized | 0.153 | 0.436 | 1.289 |
| JumpReLU | Semi-amortized | 0.194 | 0.555 | 1.668 |
| JumpReLU | Unamortized | 0.183 | 0.506 | 1.523 |
| JumpReLU | Amortized | 0.140 | 0.398 | 1.193 |
| Standard | Semi-amortized | 0.164 | 0.485 | 1.432 |
| Standard | Unamortized | 0.148 | 0.437 | 1.291 |
| Standard | Amortized | 0.151 | 0.344 | 1.027 |

Table 2: GIS results on **Pythia-160M-deduped** (`resid_post`, layer 8). $S$ is the scorer LM log-probability gain under matched KL targets $\kappa \in \{0.10, 0.33, 1.00\}$.

## D   DYNAMICS OF THE SEMI-AMORTIZED PATTERN

We further ablate the semi-amortized BatchTopK SAE by varying the number of refinement steps $T$ in the inner sparse-coding loop (Table 3). Across both Pythia-160M-deduped (layer 8) and Gemma-2-2B (layer 12), increasing $T$ monotonically reduces the reconstruction error (NMSE), while the pathological metrics remain largely stable. For Pythia, NMSE drops from 0.477 at $T = 5$ to 0.132 at $T = 25$ and 0.046 at $T = 50$, with the majority of the gains already obtained by $T \approx 20$–30. A similar trend holds for Gemma-2-2B, where NMSE decreases from 0.072 at $T = 5$ to 0.025 at $T = 25$ and 0.014 at $T = 50$. In contrast, the Dead Rate stays at zero for all settings, Dense Rate@0.1/0.2 only fluctuates within a narrow band, and the F1@1/F1@2 scores are almost flat. The feature-splitting indicator $\Delta$F1 exhibits a mild increase with $T$ on Pythia (from 0.026 at $T = 5$ to $\approx 0.06$ around $T = 20$–40) and then slightly decreases again for very large $T$, while the Absorption Rate is either nearly constant (Pythia) or decreases slightly (Gemma). Overall, these results suggest that semi-amortized inference is robust to the exact choice of refinement budget: a moderate number of refinement steps ($T \approx 20$–30) already recovers most of the reconstruction benefit of iterative inference, without materially changing the profile of pathological phenomena, and at a fraction of the cost of the fully non-amortized solver used to estimate the amortization gap.

Table 3: Pathological Phenomena Metrics of BatchTopK SAE under Semi-Amortized Patterns at Different ISTA Iteration Steps

| Iteration Step (ISTA) | NMSE | Dead Rate | Dense Rate@0.1 | Dense Rate@0.2 | F1@1 | F1@2 | $\Delta F1$ | Absorption Rate |
|---|---|---|---|---|---|---|---|---|
| **Pythia-160m-deduped, Layer 8** | | | | | | | | |
| 5 | 0.477 | 0.000 | 0.052 | 0.030 | 0.723 | 0.749 | 0.026 | 0.110 |
| 10 | 0.388 | 0.000 | 0.064 | 0.049 | 0.704 | 0.722 | 0.017 | 0.144 |
| 15 | 0.250 | 0.000 | 0.060 | 0.047 | 0.692 | 0.746 | 0.054 | 0.134 |
| 20 | 0.177 | 0.000 | 0.057 | 0.034 | 0.689 | 0.755 | 0.066 | 0.119 |
| 25 | 0.132 | 0.000 | 0.048 | 0.034 | 0.688 | 0.755 | 0.067 | 0.121 |
| 30 | 0.102 | 0.000 | 0.048 | 0.034 | 0.687 | 0.755 | 0.068 | 0.121 |
| 35 | 0.081 | 0.000 | 0.046 | 0.036 | 0.687 | 0.756 | 0.068 | 0.122 |
| 40 | 0.066 | 0.000 | 0.062 | 0.040 | 0.687 | 0.755 | 0.068 | 0.122 |
| 45 | 0.055 | 0.000 | 0.057 | 0.039 | 0.687 | 0.726 | 0.038 | 0.121 |
| 50 | 0.046 | 0.000 | 0.059 | 0.036 | 0.687 | 0.726 | 0.039 | 0.121 |
| **Gemma-2-2b, Layer 12** | | | | | | | | |
| 5 | 0.072 | 0.000 | 0.231 | 0.150 | 0.647 | 0.682 | 0.034 | 0.014 |
| 10 | 0.048 | 0.000 | 0.207 | 0.129 | 0.648 | 0.682 | 0.034 | 0.010 |
| 15 | 0.037 | 0.000 | 0.215 | 0.123 | 0.648 | 0.682 | 0.034 | 0.007 |
| 20 | 0.029 | 0.000 | 0.218 | 0.167 | 0.648 | 0.682 | 0.034 | 0.005 |
| 25 | 0.025 | 0.000 | 0.223 | 0.168 | 0.648 | 0.682 | 0.034 | 0.004 |
| 30 | 0.021 | 0.000 | 0.239 | 0.160 | 0.648 | 0.682 | 0.034 | 0.003 |
| 35 | 0.019 | 0.000 | 0.210 | 0.135 | 0.648 | 0.682 | 0.034 | 0.003 |
| 40 | 0.017 | 0.000 | 0.226 | 0.125 | 0.648 | 0.681 | 0.034 | 0.003 |
| 45 | 0.015 | 0.000 | 0.230 | 0.154 | 0.647 | 0.681 | 0.034 | 0.003 |
| 50 | 0.014 | 0.000 | 0.202 | 0.142 | 0.647 | 0.681 | 0.033 | 0.002 |

# E  LIMITATIONS

Although the metrics for many pathological phenomena in SAEs have been significantly alleviated by the semi-amortized and non-amortized encoding methods proposed in this paper, there are still some anomalies. For instance, in the Topk SAE case corresponding to Gemma-2-2b/layer12, the non-amortized approach instead exacerbates feature splitting and feature absorption, the reasons for which remain worthy of exploration (4). Additionally, since both semi-amortized and non-amortized methods involve per-sample iterative traditional sparse coding, which aligns with the goal of monosemanticity, it exacerbates the existing scalability issues of SAEs. Therefore, how to balance scalability while mitigating the limitations introduced by amortized inference remains an area worthy of researchers' exploration. Potential approaches could include meta learning proposed in VAEs (Iakovleva et al., 2020), or gradually updating sparse codes with online dictionary learning (Mairal et al., 2009).

# F  EXPERIMENTAL RESULTS

## F.1  EXPERIMENTAL RESULTS OF SECTION 4.3

## F.2  EXPERIMENTAL RESULTS OF SECTION 5

Table 4: Variation of Different Pathological Phenomena Corresponding to Standard SAE at Different Sparsity Levels from the Perspective of Training Dynamics.

| Trainer | Checkpoint | NMSE | Dead Rate | Dense Rate@0.2 | F1@1 | F1@2 | $\Delta$F1 | Absorption Rate | $\bar{\Delta}$ |
|---|---|---|---|---|---|---|---|---|---|
| 0 | 0 | 0.9413 | 0.2587 | 0.5346 | 0.7129 | 0.8051 | 0.0923 | 0.0186 | 25978.81 |
| 0 | 244 | 0.1555 | 0.2368 | 0.6643 | 0.7704 | 0.8240 | 0.0535 | 0.0694 | 4350.55 |
| 0 | 2441 | 0.0332 | 0.2300 | 0.7397 | 0.4534 | 0.6164 | 0.1631 | 0.4564 | 855.77 |
| 0 | 24414 | 0.1048 | 0.2214 | 0.0862 | 0.6034 | 0.6383 | 0.0349 | 0.2342 | 1689.17 |
| 0 | 772 | 0.0303 | 0.2313 | 0.7295 | 0.6356 | 0.6864 | 0.0508 | 0.2554 | 880.45 |
| 0 | 7720 | 0.0764 | 0.2285 | 0.0438 | 0.5949 | 0.6509 | 0.0559 | 0.2546 | 1449.20 |
| 0 | 77203 | 0.1174 | 0.2185 | 0.0890 | 0.5590 | 0.7156 | 0.1566 | 0.4352 | 1773.85 |
| 1 | 0 | 0.9413 | 0.2587 | 0.5346 | 0.7129 | 0.8051 | 0.0923 | 0.0186 | 25978.70 |
| 1 | 244 | 0.1550 | 0.2368 | 0.6642 | 0.7701 | 0.8240 | 0.0539 | 0.0728 | 4335.57 |
| 1 | 2441 | 0.0528 | 0.2299 | 0.7250 | 0.6075 | 0.6440 | 0.0366 | 0.1686 | 1419.51 |
| 1 | 24414 | 0.1238 | 0.2224 | 0.0502 | 0.5335 | 0.6275 | 0.0940 | 0.5034 | 1748.02 |
| 1 | 772 | 0.0316 | 0.2316 | 0.7287 | 0.6316 | 0.6938 | 0.0622 | 0.2698 | 913.97 |
| 1 | 7720 | 0.0858 | 0.2282 | 0.0159 | 0.6231 | 0.6437 | 0.0206 | 0.0398 | 1606.65 |
| 1 | 77203 | 0.1430 | 0.2224 | 0.0535 | 0.5746 | 0.6801 | 0.1055 | 0.4900 | 1996.48 |
| 2 | 0 | 0.9413 | 0.2587 | 0.5346 | 0.7129 | 0.8051 | 0.0923 | 0.0186 | 25978.86 |
| 2 | 244 | 0.1544 | 0.2368 | 0.6647 | 0.7685 | 0.8239 | 0.0554 | 0.0796 | 4316.75 |
| 2 | 2441 | 0.0964 | 0.2301 | 0.6917 | 0.4788 | 0.5679 | 0.0891 | 0.4530 | 2603.39 |
| 2 | 24414 | 0.1318 | 0.2294 | 0.0287 | 0.5129 | 0.6418 | 0.1289 | 0.5222 | 1325.94 |
| 2 | 772 | 0.0348 | 0.2318 | 0.7286 | 0.6250 | 0.6918 | 0.0668 | 0.3008 | 999.66 |
| 2 | 7720 | 0.0963 | 0.2289 | 0.0070 | 0.5082 | 0.6457 | 0.1375 | 0.4992 | 1763.15 |
| 2 | 77203 | 0.1533 | 0.2361 | 0.0286 | 0.5831 | 0.6342 | 0.0511 | 0.3398 | 1669.68 |
| 3 | 0 | 0.9413 | 0.2587 | 0.5346 | 0.7129 | 0.8051 | 0.0923 | 0.0186 | 25978.76 |
| 3 | 244 | 0.1537 | 0.2368 | 0.6655 | 0.7683 | 0.8241 | 0.0557 | 0.1076 | 4293.68 |
| 3 | 2441 | 0.2042 | 0.2296 | 0.6129 | 0.5224 | 0.6094 | 0.0869 | 0.5208 | 5528.09 |
| 3 | 24414 | 0.1294 | 0.2595 | 0.0104 | 0.5998 | 0.6313 | 0.0315 | 0.2012 | 306.95 |
| 3 | 772 | 0.0427 | 0.2317 | 0.7267 | 0.6108 | 0.6803 | 0.0695 | 0.3520 | 1203.42 |
| 3 | 7720 | 0.1114 | 0.2300 | 0.0044 | 0.5248 | 0.6513 | 0.1265 | 0.4212 | 1638.43 |
| 3 | 77203 | 0.1395 | 0.2686 | 0.0128 | 0.5334 | 0.6350 | 0.1016 | 0.3860 | 212.58 |
| 4 | 0 | 0.9413 | 0.2587 | 0.5346 | 0.7129 | 0.8051 | 0.0923 | 0.0186 | 25978.76 |
| 4 | 244 | 0.1535 | 0.2367 | 0.6663 | 0.7655 | 0.8212 | 0.0557 | 0.1148 | 4284.31 |
| 4 | 2441 | 0.3266 | 0.2288 | 0.5286 | 0.4851 | 0.6023 | 0.1171 | 0.5840 | 8849.26 |
| 4 | 24414 | 0.1402 | 0.2838 | 0.0045 | 0.5075 | 0.6299 | 0.1224 | 0.4498 | 92.56 |
| 4 | 772 | 0.0544 | 0.2319 | 0.7234 | 0.5964 | 0.6623 | 0.0659 | 0.3838 | 1507.16 |
| 4 | 7720 | 0.1255 | 0.2336 | 0.0039 | 0.6571 | 0.6635 | 0.0064 | 0.0794 | 1419.27 |
| 4 | 77203 | 0.1388 | 0.3023 | 0.0074 | 0.5646 | 0.6626 | 0.0980 | 0.4772 | -388.11 |
| 5 | 0 | 0.9413 | 0.2587 | 0.5346 | 0.7129 | 0.8051 | 0.0923 | 0.0186 | 25978.77 |
| 5 | 244 | 0.1537 | 0.2367 | 0.6669 | 0.7570 | 0.7937 | 0.0367 | 0.1234 | 4285.21 |
| 5 | 2441 | 0.5447 | 0.2278 | 0.4059 | 0.5490 | 0.6196 | 0.0705 | 0.3480 | 14650.23 |
| 5 | 24414 | 0.1633 | 0.3221 | 0.0013 | 0.5146 | 0.5589 | 0.0443 | 0.1422 | -161.76 |
| 5 | 772 | 0.0833 | 0.2319 | 0.7186 | 0.6006 | 0.6431 | 0.0425 | 0.2176 | 2259.97 |
| 5 | 7720 | 0.1549 | 0.2297 | 0.0034 | 0.6638 | 0.6797 | 0.0160 | 0.0032 | 1483.02 |
| 5 | 77203 | 0.1537 | 0.3524 | 0.0028 | 0.1398 | 0.6406 | 0.5008 | 0.9164 | -782.56 |

Table 5: Variation of Different Pathological Phenomena Corresponding to Top-k SAE at Different Sparsity Levels from the Perspective of Training Dynamics.

| Trainer | Checkpoint | NMSE | Dead Rate | Dense Rate@0.2 | F1@1 | F1@2 | ΔF1 | Absorption Rate | $\bar{\Delta}$ |
|---|---|---|---|---|---|---|---|---|---|
| 0 | 0 | 0.9413 | 0.2587 | 0.5346 | 0.7129 | 0.8051 | 0.0923 | 0.0186 | 25978.81 |
| 0 | 244 | 0.1555 | 0.2368 | 0.6643 | 0.7704 | 0.8240 | 0.0535 | 0.0694 | 4350.55 |
| 0 | 2441 | 0.0332 | 0.2300 | 0.7397 | 0.4534 | 0.6164 | 0.1631 | 0.4564 | 855.77 |
| 0 | 24414 | 0.1048 | 0.2214 | 0.0862 | 0.6034 | 0.6383 | 0.0349 | 0.2342 | 1689.17 |
| 0 | 772 | 0.0303 | 0.2313 | 0.7295 | 0.6356 | 0.6864 | 0.0508 | 0.2554 | 880.45 |
| 0 | 7720 | 0.0764 | 0.2285 | 0.0438 | 0.5949 | 0.6509 | 0.0559 | 0.2546 | 1449.20 |
| 0 | 77203 | 0.1174 | 0.2185 | 0.0890 | 0.5590 | 0.7156 | 0.1566 | 0.4352 | 1773.85 |
| 1 | 0 | 0.9413 | 0.2587 | 0.5346 | 0.7129 | 0.8051 | 0.0923 | 0.0186 | 25978.70 |
| 1 | 244 | 0.1550 | 0.2368 | 0.6642 | 0.7701 | 0.8240 | 0.0539 | 0.0728 | 4335.57 |
| 1 | 2441 | 0.0528 | 0.2299 | 0.7250 | 0.6075 | 0.6440 | 0.0366 | 0.1686 | 1419.51 |
| 1 | 24414 | 0.1238 | 0.2224 | 0.0502 | 0.5335 | 0.6275 | 0.0940 | 0.5034 | 1748.02 |
| 1 | 772 | 0.0316 | 0.2316 | 0.7287 | 0.6316 | 0.6938 | 0.0622 | 0.2698 | 913.97 |
| 1 | 7720 | 0.0858 | 0.2282 | 0.0159 | 0.6231 | 0.6437 | 0.0206 | 0.0398 | 1606.65 |
| 1 | 77203 | 0.1430 | 0.2224 | 0.0535 | 0.5746 | 0.6801 | 0.1055 | 0.4900 | 1996.48 |
| 2 | 0 | 0.9413 | 0.2587 | 0.5346 | 0.7129 | 0.8051 | 0.0923 | 0.0186 | 25978.86 |
| 2 | 244 | 0.1544 | 0.2368 | 0.6647 | 0.7685 | 0.8239 | 0.0554 | 0.0796 | 4316.75 |
| 2 | 2441 | 0.0964 | 0.2301 | 0.6917 | 0.4788 | 0.5679 | 0.0891 | 0.4530 | 2603.39 |
| 2 | 24414 | 0.1318 | 0.2294 | 0.0287 | 0.5129 | 0.6418 | 0.1289 | 0.5222 | 1325.94 |
| 2 | 772 | 0.0348 | 0.2318 | 0.7286 | 0.6250 | 0.6918 | 0.0668 | 0.3008 | 999.66 |
| 2 | 7720 | 0.0963 | 0.2289 | 0.0070 | 0.5082 | 0.6457 | 0.1375 | 0.4992 | 1763.15 |
| 2 | 77203 | 0.1533 | 0.2361 | 0.0286 | 0.5831 | 0.6342 | 0.0511 | 0.3398 | 1669.68 |
| 3 | 0 | 0.9413 | 0.2587 | 0.5346 | 0.7129 | 0.8051 | 0.0923 | 0.0186 | 25978.76 |
| 3 | 244 | 0.1537 | 0.2368 | 0.6655 | 0.7683 | 0.8241 | 0.0557 | 0.1076 | 4293.68 |
| 3 | 2441 | 0.2042 | 0.2296 | 0.6129 | 0.5224 | 0.6094 | 0.0869 | 0.5208 | 5528.09 |
| 3 | 24414 | 0.1294 | 0.2595 | 0.0104 | 0.5998 | 0.6313 | 0.0315 | 0.2012 | 306.95 |
| 3 | 772 | 0.0427 | 0.2317 | 0.7267 | 0.6108 | 0.6803 | 0.0695 | 0.3520 | 1203.42 |
| 3 | 7720 | 0.1114 | 0.2300 | 0.0044 | 0.5248 | 0.6513 | 0.1265 | 0.4212 | 1638.43 |
| 3 | 77203 | 0.1395 | 0.2686 | 0.0128 | 0.5334 | 0.6350 | 0.1016 | 0.3860 | 212.58 |
| 4 | 0 | 0.9413 | 0.2587 | 0.5346 | 0.7129 | 0.8051 | 0.0923 | 0.0186 | 25978.76 |
| 4 | 244 | 0.1535 | 0.2367 | 0.6663 | 0.7655 | 0.8212 | 0.0557 | 0.1148 | 4284.31 |
| 4 | 2441 | 0.3266 | 0.2288 | 0.5286 | 0.4851 | 0.6023 | 0.1171 | 0.5840 | 8849.26 |
| 4 | 24414 | 0.1402 | 0.2838 | 0.0045 | 0.5075 | 0.6299 | 0.1224 | 0.4498 | 92.56 |
| 4 | 772 | 0.0544 | 0.2319 | 0.7234 | 0.5964 | 0.6623 | 0.0659 | 0.3838 | 1507.16 |
| 4 | 7720 | 0.1255 | 0.2336 | 0.0039 | 0.6571 | 0.6635 | 0.0064 | 0.0794 | 1419.27 |
| 4 | 77203 | 0.1388 | 0.3023 | 0.0074 | 0.5646 | 0.6626 | 0.0980 | 0.4772 | -388.11 |
| 5 | 0 | 0.9413 | 0.2587 | 0.5346 | 0.7129 | 0.8051 | 0.0923 | 0.0186 | 25978.77 |
| 5 | 244 | 0.1537 | 0.2367 | 0.6669 | 0.7570 | 0.7937 | 0.0367 | 0.1234 | 4285.21 |
| 5 | 2441 | 0.5447 | 0.2278 | 0.4059 | 0.5490 | 0.6196 | 0.0705 | 0.3480 | 14650.23 |
| 5 | 24414 | 0.1633 | 0.3221 | 0.0013 | 0.5146 | 0.5589 | 0.0443 | 0.1422 | -161.76 |
| 5 | 772 | 0.0833 | 0.2319 | 0.7186 | 0.6006 | 0.6431 | 0.0425 | 0.2176 | 2259.97 |
| 5 | 7720 | 0.1549 | 0.2297 | 0.0034 | 0.6638 | 0.6797 | 0.0160 | 0.0032 | 1483.02 |
| 5 | 77203 | 0.1537 | 0.3524 | 0.0028 | 0.1398 | 0.6406 | 0.5008 | 0.9164 | -782.56 |

Table 6: Pathological Phenomenon Metrics for Different SAE Variants under Various Amortization Patterns

| SAE Variants | Pattern | NMSE | Dead Rate | Dense Rate@0.1 | Dense Rate@0.2 | F1@1 | F1@2 | $\Delta F1$ | Absorption Rate |
|---|---|---|---|---|---|---|---|---|---|
| **Pythia-160m-deduped, Layer 8** | | | | | | | | | |
| Standard | Full-Amortized | 0.109 | 0.295 | 0.093 | 0.046 | 0.606 | 0.649 | 0.043 | 0.144 |
| | Semi-Amortized | 0.078 | 0.260 | 0.095 | 0.043 | 0.600 | 0.641 | 0.041 | 0.158 |
| | Non-Amortized | 0.107 | 0.338 | 0.094 | 0.045 | 0.590 | 0.637 | 0.047 | 0.283 |
| GatedSAE | Full-Amortized | 1.947 | 0.001 | 0.612 | 0.432 | 0.655 | 0.692 | 0.037 | 0.210 |
| | Semi-Amortized | 0.014 | 0.001 | 0.615 | 0.431 | 0.655 | 0.690 | 0.035 | 0.161 |
| | Non-Amortized | 0.000 | 0.008 | 0.610 | 0.467 | 0.676 | 0.671 | -0.005 | 0.064 |
| JumpRelu | Full-Amortized | 0.755 | 0.371 | 0.032 | 0.011 | 0.673 | 0.742 | 0.070 | 0.016 |
| | Semi-Amortized | 0.255 | 0.321 | 0.033 | 0.017 | 0.670 | 0.744 | 0.074 | 0.143 |
| | Non-Amortized | 0.113 | 0.055 | 0.033 | 0.009 | 0.672 | 0.681 | 0.009 | 0.066 |
| TopK | Full-Amortized | 1.499 | 0.307 | 0.028 | 0.015 | 0.626 | 0.679 | 0.053 | 0.225 |
| | Semi-Amortized | 0.087 | 0.022 | 0.029 | 0.010 | 0.648 | 0.683 | 0.036 | 0.134 |
| | Non-Amortized | 0.242 | 0.012 | 0.029 | 0.008 | 0.535 | 0.558 | 0.023 | 0.452 |
| BatchTopK | Full-Amortized | 0.537 | 0.316 | 0.053 | 0.040 | 0.700 | 0.748 | 0.047 | 0.171 |
| | Semi-Amortized | 0.101 | 0.000 | 0.055 | 0.045 | 0.687 | 0.755 | 0.031 | 0.121 |
| | Non-Amortized | 0.017 | 0.000 | 0.053 | 0.043 | 0.726 | 0.758 | 0.031 | 0.033 |
| Matryoshka | Full-Amortized | 0.369 | 0.219 | 0.107 | 0.047 | 0.689 | 0.722 | 0.033 | 0.240 |
| | Semi-Amortized | 0.033 | 0.000 | 0.121 | 0.055 | 0.681 | 0.716 | 0.035 | 0.219 |
| | Non-Amortized | 0.011 | 0.007 | 0.114 | 0.049 | 0.677 | 0.729 | 0.052 | 0.172 |
| LocA-SAE | Loc-Amortized | 0.427 | 0.000 | 0.079 | 0.044 | 0.714 | 0.728 | 0.013 | 0.055 |
| **Gemma-2-2b, Layer 12** | | | | | | | | | |
| Standard | Full-Amortized | 0.193 | 0.221 | 0.251 | 0.108 | 0.722 | 0.737 | 0.015 | 0.010 |
| | Semi-Amortized | 0.114 | 0.212 | 0.254 | 0.072 | 0.716 | 0.737 | 0.021 | 0.007 |
| | Non-Amortized | 0.135 | 0.208 | 0.255 | 0.084 | 0.690 | 0.743 | 0.053 | 0.011 |
| GatedSAE | Full-Amortized | 1.580 | 0.000 | 0.950 | 0.854 | 0.882 | 0.882 | 0.000 | 0.000 |
| | Semi-Amortized | 0.013 | 0.000 | 0.951 | 0.831 | 0.879 | 0.881 | 0.001 | 0.000 |
| | Non-Amortized | 0.001 | 0.000 | 0.939 | 0.879 | 0.851 | 0.854 | 0.003 | 0.000 |
| JumpRelu | Full-Amortized | 0.341 | 0.102 | 0.056 | 0.018 | 0.815 | 0.847 | 0.032 | 0.923 |
| | Semi-Amortized | 0.236 | 0.086 | 0.057 | 0.020 | 0.841 | 0.842 | 0.001 | 0.000 |
| | Non-Amortized | 0.225 | 0.028 | 0.059 | 0.014 | 0.838 | 0.848 | 0.010 | 0.000 |
| TopK | Full-Amortized | 0.882 | 0.123 | 0.051 | 0.021 | 0.838 | 0.838 | 0.000 | 0.000 |
| | Semi-Amortized | 0.186 | 0.011 | 0.052 | 0.018 | 0.836 | 0.839 | 0.004 | 0.000 |
| | Non-Amortized | 0.274 | 0.011 | 0.053 | 0.013 | 0.571 | 0.768 | 0.197 | 0.400 |
| BatchTopK | Full-Amortized | 0.152 | 0.136 | 0.231 | 0.147 | 0.646 | 0.708 | 0.062 | 0.093 |
| | Semi-Amortized | 0.021 | 0.000 | 0.230 | 0.150 | 0.648 | 0.682 | 0.034 | 0.003 |
| | Non-Amortized | 0.005 | 0.000 | 0.235 | 0.154 | 0.629 | 0.669 | 0.040 | 0.018 |
| Matryoshka | Full-Amortized | 0.573 | 0.113 | 0.077 | 0.037 | 0.690 | 0.681 | -0.008 | 0.732 |
| | Semi-Amortized | 0.059 | 0.000 | 0.083 | 0.043 | 0.685 | 0.678 | -0.006 | 0.431 |
| | Non-Amortized | 0.012 | 0.000 | 0.083 | 0.040 | 0.613 | 0.654 | 0.041 | 0.020 |
| LocA-SAE | Loc-Amortized | 0.211 | 0.000 | 0.017 | 0.066 | 0.670 | 0.694 | 0.024 | 0.023 |

# G    EXPERIMENTAL DETAILS

## G.1    EXPERIMENTAL DETAILS OF SECTION 5

---

**Algorithm 1** Implementation Flow for the Experiment in Section 6

**Require:** models $\mathcal{M}$, variants $\mathcal{V}$, corpus $\mathcal{C}$, layer $\ell$, token budget $N=10{,}000$, encoder $(W_{\text{enc}}, b_{\text{enc}})$, decoder $D$, base $\lambda$, target density $T=0.1$, tolerance $\varepsilon$

1: **for** $(m, v) \in \mathcal{M} \times \mathcal{V}$ **do**
2:     $X \leftarrow \text{COLLECTHIDDENSTATES}(m, \mathcal{C}, \ell, N)$        ▷ first $N$ tokens → layer-$\ell$ activations
3:     $z^{\text{full}} \leftarrow \text{ENC}_{\text{full}}(X; W_{\text{enc}}, b_{\text{enc}}, v)$
4:     $t \leftarrow \text{DENSE@}0.1(z^{\text{full}})$
5:     $s_{\text{semi}} \leftarrow \text{CALIBRATE}(\text{Semi}, t, \varepsilon); \quad s_{\text{non}} \leftarrow \text{CALIBRATE}(\text{Non}, t, \varepsilon)$
6:     $z^{\text{semi}} \leftarrow \text{PGD}(X, D, \lambda s_{\text{semi}}; \text{init} = z^{\text{full}}, T = 30)$
7:     $z^{\text{non}} \leftarrow \text{ISTA}(X, D, \lambda s_{\text{non}}; \text{init} = 0, T = 200)$
8:     $\hat{X}^r \leftarrow z^r D^\top$ for $r \in \{\text{full}, \text{semi}, \text{non}\}$
9:     $\text{EVALUATE}(\{\hat{X}^r\}, \{z^r\})$        ▷ NMSE, Dead, DENSE@0.1/0.2, $F1@1$, $F1@2$, $\Delta F1$, Absorption

---

## H  CONFIGURATION INFORMATION

Table 7: Key hyperparameter configurations for six different SAE architectures. All SAEs were trained on the residual stream of layer 12 of the Gemma-2-2B model, with an activation dimension of 2304.

| Parameter | Gated SAE | JumpReLU SAE | Standard SAE | Top-K SAE | BatchTopK SAE | Matryoshka SAE |
|---|---|---|---|---|---|---|
| Dictionary Size ($d_{\text{dict}}$) | 16384 ($2^{14}$) | 16384 ($2^{14}$) | 16384 ($2^{14}$) | 16384 ($2^{14}$) | 16384 ($2^{14}$) | 16384 ($2^{14}$) |
| Learning Rate (LR) | $3 \times 10^{-4}$ | $3 \times 10^{-4}$ | $3 \times 10^{-4}$ | $3 \times 10^{-4}$ | $3 \times 10^{-4}$ | $3 \times 10^{-4}$ |
| *Sparsity-Related Parameters* | | | | | | |
| $L_1$ Penalty | 0.012 | — | 0.012 | — | — | — |
| Sparsity Penalty | — | 1.0 | — | — | — | — |
| Target $L_0$ | — | 20 | — | — | — | — |
| Top-K Value ($k$) | — | — | — | 20 | 320 | 20 |
| *Training Strategy Parameters* | | | | | | |
| LR Warmup Steps | 1000 | N/A | 1000 | 1000 | 1000 | 1000 |
| Sparsity Warmup Steps | 5000 | 5000 | 5000 | N/A | N/A | N/A |

Table 8: Key hyperparameter configurations for six different SAE architectures trained on the EleutherAI/pythia-160m-deduped model. All SAEs were trained on the residual stream of layer 8, with an activation dimension of 768.

| Parameter | Gated SAE | JumpReLU SAE | Standard SAE | Top-K SAE | BatchTopK SAE | Matryoshka SAE |
|---|---|---|---|---|---|---|
| Dictionary Size ($d_{\text{dict}}$) | 16384 ($2^{14}$) | 16384 ($2^{14}$) | 16384 ($2^{14}$) | 16384 ($2^{14}$) | 16384 ($2^{14}$) | 16384 ($2^{14}$) |
| Learning Rate (LR) | $3 \times 10^{-4}$ | $3 \times 10^{-4}$ | $3 \times 10^{-4}$ | $3 \times 10^{-4}$ | $3 \times 10^{-4}$ | $3 \times 10^{-4}$ |
| *Sparsity-Related Parameters* | | | | | | |
| $L_1$ Penalty | 0.012 | — | 0.012 | — | — | — |
| Sparsity Penalty | — | 1.0 | — | — | — | — |
| Target $L_0$ | — | 20 | — | — | — | — |
| Top-K Value ($k$) | — | — | — | 20 | 20 | 20 |
| *Training Strategy Parameters* | | | | | | |
| LR Warmup Steps | 1000 | N/A | 1000 | 1000 | 1000 | 1000 |
| Sparsity Warmup Steps | 5000 | 5000 | 5000 | N/A | N/A | N/A |

## I  EVALUATION METRICS

Table 9: Evaluation Metrics ($Z_{\mathrm{relu}} = \max(Z, 0)$; feature ranking uses $Z_s$ which z-score normalizes $Z_{\mathrm{relu}}$ per column).

| Variable Name | Meaning | Formula |
|---|---|---|
| NMSE | Normalized mean squared reconstruction error. | $\dfrac{\mathbb{E}\big[\|x - \hat{x}\|_2^2\big]}{\mathbb{E}\big[\|x\|_2^2\big] + \varepsilon}$, where $\varepsilon = 10^{-9}$, $\hat{x} = z\,\mathbf{D}^\top$. |
| Dead Rate | Fraction of dead latents. | $\dfrac{1}{M}\sum_{j=1}^{M}\mathbb{I}\big(\mathrm{freq}_j \leq \theta\big)$, where $\theta = 10^{-6}$, $\mathrm{freq}_j = \Pr\big((Z_{\mathrm{relu}})_{:,j} > 0\big)$. |
| Dense Rate@0.2 | Fraction of latents firing at least 20% of tokens (more frequently active). | $\dfrac{1}{M}\sum_{j=1}^{M}\mathbb{I}\big(\mathrm{freq}_j \geq 0.2\big)$. |
| F1@1 | F1 score of a linear probe using the top-1 ranked latent (by $|\mathrm{corr}(Z_{s,\cdot j}, y)|$). | $\mathrm{F1@1} = \mathrm{F1}\big(\mathrm{LR}(Z_s[:, \mathrm{order}[0]] \to y)\big)$. |
| F1@2 | F1 score of a linear probe using the top-2 ranked latents. | $\mathrm{F1@2} = \mathrm{F1}\big(\mathrm{LR}(Z_s[:, \mathrm{order}[:2]] \to y)\big)$. |
| $\Delta$F1 | Marginal improvement from 1 to 2 features; larger values indicate stronger feature splitting. | $\Delta\mathrm{F1} = \mathrm{F1@2} - \mathrm{F1@1}$. |
| Absorption Rate | On positive-label tokens, fraction where the dominant latent is inactive while any of the next top-$K$ latents is active. | Let $m = \mathrm{order}[0]$, $A = \mathrm{order}[1 : 1+K]$ (default $K = 5$). $\mathrm{Absorb} = \dfrac{1}{N_+}\sum_{i:\,y_i=1}\mathbb{I}\Big((Z_{\mathrm{relu}})_{i,m} \leq 0 \,\wedge\, \max_{j \in A}(Z_{\mathrm{relu}})_{i,j} > 0\Big)$. |
| Amortization Gap | Suboptimality of amortized codes vs. per-token L1 solution (ISTA-200). | $\mathrm{Gap} = \mathbb{E}\big[L(z^{\mathrm{amort}}) - L(z^\star)\big]$, $L(z) = \frac{1}{2}\|x - z\mathbf{D}^\top\|_2^2 + \lambda\|z\|_1$, $z^\star \approx \mathrm{ISTA}_{200}(x; \mathbf{D}, \lambda)$. |

**Notes.** (1) Labels $y$ use a norm-threshold heuristic unless otherwise stated: $y_i = \mathbb{I}(\|x_i\|_2 > \mathrm{median}_i\|x_i\|_2)$. (2) Linear probes are trained on $Z_s$ with class_weight=balanced; ranking uses $|\mathrm{corr}(Z_{s,\cdot j}, y)|$. (3) For Top-$K$ SAEs, selection is applied first, then $Z_{\mathrm{relu}}$ is used for firing-based metrics. (4) When comparing amortized/semi-amortized/ISTA codes, $\lambda$ can be calibrated to match a target density.

