# OpenReview forum: "The  Price of Amortized inference in Sparse Autoencoders"
_ICLR.cc/2026/Conference — ICLR 2026 Poster_

### Official Review · Reviewer_gX9Z · 2025-10-30

**Soundness:** 3
**Presentation:** 3
**Contribution:** 3
**Rating:** 6
**Confidence:** 3

**Summary:**

This paper investigates fundamental limitations of Sparse Autoencoders (SAEs) for mechanistic interpretability, arguing that the amortized inference paradigm conflicts with the instance-level optimality required for monosemantic features. The authors demonstrate through training dynamics analysis that pathological phenomena (feature absorption, splitting, dead/dense latents) are interconnected trade-offs arising from parameter sharing under global reconstruction-sparsity constraints. They propose semi-amortized and non-amortized encoding methods as alternatives, showing improved reconstruction, reduced dead latents, and better performance on downstream intervention tasks.

**Strengths:**

1. The paper articulates a fundamental issue that has been under-recognized in the SAE literature - that global optimization objectives may be misaligned with the goals of interpretability
2. The training dynamics experiments across multiple checkpoints, sparsity levels, and SAE variants provide robust evidence for the claims
3. The amortization gap perspective from variational inference provides a principled lens for understanding SAE limitations
4. The downstream task experiments (TPP, GIS) demonstrate that the improvements translate to better interpretability and controllability

**Weaknesses:**

1. The computational cost of non-amortized inference (200 ISTA iterations × N samples) is mentioned as a limitation but needs more serious treatment. For large-scale interpretability applications, this may be prohibitive. The paper should provide concrete runtime comparisons and discuss when the trade-off is worthwhile.

2. While the paper shows that pathological phenomena are correlated, it doesn't fully explain why instance-level optimality necessarily leads to better monosemanticity. The connection between reconstruction quality and semantic purity is assumed rather than proven.
3. Semi-amortized inference shows promise but receives less attention than it deserves. More systematic exploration of the trade-off between number of refinement steps and performance could be valuable.
4. The paper acknowledges some anomalous results (e.g., TopK with Gemma showing worse absorption in non-amortized setting) but doesn't investigate them thoroughly. This suggests the full picture may be more complex.

**Questions:**

1. Can you provide theoretical results or additional empirical evidence connecting instance-level reconstruction optimality to monosemanticity? What properties of the data distribution determine whether this connection holds?
2. For the outlier cases (e.g., Gemma-2-2b/L12 TopK absorption), can you provide analysis of what's happening? Does this suggest limitations of the non-amortized approach?
3. Have you explored whether better training procedures (e.g., curriculum learning, better initialization) for amortized SAEs could narrow the gap?

---

> ### Author Response · Authors · 2025-11-21
>
> ## Response to W1 and Q3:
> We acknowledge the reviewer’s concern regarding the prohibitive computational cost of ISTA-based non-amortized baselines. In the revised manuscript, we treat semi-amortized and non-amortized inference as interventions and ablation studies (Sec. 5.3). Furthermore, in Section 5.1 we introduce **LocA-SAE** (Locally Amortized Sparse Autoencoder), a practical SAE variant based on local amortization.
>
> LocA-SAE retains a single globally shared dictionary but partitions the latent variables into groups according to the angular variance of their activation directions across the dataset. Latents exhibiting similar degrees of polysemanticity are assigned to the same group; each group is then equipped with its own dedicated lightweight encoder and receives an independent sparsity penalty. Angular variance serves as a proxy for the diversity of activation patterns supported by a given latent and thus indirectly measures its degree of polysemanticity. By grouping latents with comparable polysemanticity and applying group-specific encoding and sparsity regularization, LocA-SAE largely eliminates the unfair competition that arises under global reconstruction–sparsity constraints when latents of widely differing polysemanticity are forced to share the same encoder and regularization strength.
>
> The detailed methodology and corresponding experimental results have been incorporated into Section 5 of the revised manuscript. Preliminary results demonstrate that LocA-SAE substantially mitigates pathological phenomena such as feature splitting and feature absorption with only a minor degradation in reconstruction performance, and even completely eliminates dead latent in all cases.
> ## Response to W2 and Q1:
> We acknowledge the reviewer's concern regarding the correlation between instance-level optimality and improved monosemanticity. In the revised manuscript, we have framed the original semi-amortized and non-amortized settings as intervention and ablation studies, designed to empirically demonstrate the relationship between instance-level optimality and pathological metrics. We observe that as the reliance on full amortization is progressively reduced, pathological metrics are mitigated across most SAE variants, accompanied by improved performance on downstream tasks such as TPP and GIS. These results provide strong empirical support for the link between instance-level optimality and enhanced monosemanticity.
>
> Regarding the reviewer’s question on data distribution properties, we identify the structure of the activation distribution as the key determinant. The connection between instance-level reconstruction optimality and monosemanticity is most pronounced when representations are effectively approximated by a sparse superposition of distinct directions. In this regime, each example activates only a few factors, rendering the instance-optimal sparse code essentially identifiable [1]. Furthermore, in long-tailed or multimodal distributions containing rare or context-specific concepts, a globally amortized encoder tends to underfit these rare directions in favor of frequent ones. Consequently, pushing inference closer to the instance-optimal solution reliably recovers these features and enhances monosemanticity. Conversely, when activations are dense, highly entangled, or exhibit balanced concept frequencies, the identifiability assumptions weaken. In such cases, amortization is less detrimental, resulting in a weaker empirical link between instance-optimal reconstruction and monosemanticity.
> [1] On the Theoretical Understanding of Identifiable Sparse Autoencoders and Beyond. (ArXiv)

---

> ### Author Response · Authors · 2025-11-21
>
> ## Response to W3:
> Compared to the enormous computational cost of fully non-amortized inference, semi-amortized inference does exhibit a certain trade-off between computational cost and pathological metrics. The computational cost is primarily dominated by the sample size; as the sample count increases, even a semi-amortized approach with relatively few iterations becomes difficult to scale. Therefore, the original version of this work indeed lacked discussion on the number of iteration steps. In the revised manuscript, taking BatchTopK SAE as an example, we investigate how various pathological phenomena change as the number of ISTA iteration steps increases. The detailed discussion and experimental results can be found in Appendix F of the paper. Specifically, we observe that NMSE decreases monotonically as T increases, but with clear diminishing returns: on Pythia, for example, NMSE drops from 0.477 at T=5 to 0.177 at T=20 and 0.132 at T=25, while further increasing T to 50 only reduces NMSE to 0.046. The same pattern appears on Gemma-2-2B (0.072 → 0.029 → 0.025 → 0.014 for T=5,20,25,50). In contrast, the pathological metrics that we focus, like dead rate, dense rate, feature splitting (ΔF1), and absorption are largely insensitive to T: dead latents remain at 0 for all settings, dense rates fluctuate only slightly, ΔF1 changes modestly (e.g., from 0.026 to ≈0.06 on Pythia 160m), and the absorption rate stays roughly constant on Pythia and even decreases on Gemma-2-2b.
>
> These results indicate that semi-amortized inference is quite robust to the exact refinement budget: a moderate number of steps (T≈20–30), which is much cheaper than the fully non-amortized solver used to estimate the amortization gap (200 iterations), already recovers most of the reconstruction improvement without materially changing the profile of pathological phenomena. We have updated the paper to report and discuss these findings.
> | Model | ISTA Step | NMSE | Dead Rate | Dense Rate@0.1 | Dense Rate@0.2 | F1@1 | F1@2 | $\Delta F1$ | Abs. Rate |
> | :--- | :--- | :--- | :--- | :--- | :--- | :--- | :--- | :--- | :--- |
> | **Pythia-160m**<br>(Layer 8) | 5 | 0.477 | 0.000 | 0.052 | 0.030 | 0.723 | 0.749 | 0.026 | 0.110 |
> | | 10 | 0.388 | 0.000 | 0.064 | 0.049 | 0.704 | 0.722 | 0.017 | 0.144 |
> | | 15 | 0.250 | 0.000 | 0.060 | 0.047 | 0.692 | 0.746 | 0.054 | 0.134 |
> | | 20 | 0.177 | 0.000 | 0.057 | 0.034 | 0.689 | 0.755 | 0.066 | 0.119 |
> | | 25 | 0.132 | 0.000 | 0.048 | 0.034 | 0.688 | 0.755 | 0.067 | 0.121 |
> | | 30 | 0.102 | 0.000 | 0.048 | 0.034 | 0.687 | 0.755 | 0.068 | 0.121 |
> | | 35 | 0.081 | 0.000 | 0.046 | 0.036 | 0.687 | 0.756 | 0.068 | 0.122 |
> | | 40 | 0.066 | 0.000 | 0.062 | 0.040 | 0.687 | 0.755 | 0.068 | 0.122 |
> | | 45 | 0.055 | 0.000 | 0.057 | 0.039 | 0.687 | 0.726 | 0.038 | 0.121 |
> | | 50 | 0.046 | 0.000 | 0.059 | 0.036 | 0.687 | 0.726 | 0.039 | 0.121 |
> | **Gemma-2-2b**<br>(Layer 12) | 5 | 0.072 | 0.000 | 0.231 | 0.150 | 0.647 | 0.682 | 0.034 | 0.014 |
> | | 10 | 0.048 | 0.000 | 0.207 | 0.129 | 0.648 | 0.682 | 0.034 | 0.010 |
> | | 15 | 0.037 | 0.000 | 0.215 | 0.123 | 0.648 | 0.682 | 0.034 | 0.007 |
> | | 20 | 0.029 | 0.000 | 0.218 | 0.167 | 0.648 | 0.682 | 0.034 | 0.005 |
> | | 25 | 0.025 | 0.000 | 0.223 | 0.168 | 0.648 | 0.682 | 0.034 | 0.004 |
> | | 30 | 0.021 | 0.000 | 0.239 | 0.160 | 0.648 | 0.682 | 0.034 | 0.003 |
> | | 35 | 0.019 | 0.000 | 0.210 | 0.135 | 0.648 | 0.682 | 0.034 | 0.003 |
> | | 40 | 0.017 | 0.000 | 0.226 | 0.125 | 0.648 | 0.681 | 0.034 | 0.003 |
> | | 45 | 0.015 | 0.000 | 0.230 | 0.154 | 0.647 | 0.681 | 0.034 | 0.003 |
> | | 50 | 0.014 | 0.000 | 0.202 | 0.142 | 0.647 | 0.681 | 0.033 | 0.002 |
>
> ## Response to W4 and Q2:
> We appreciate the reviewer’s careful reading and agree that, for TopK SAEs trained on Gemma-2-2b, our non-amortized baseline (ISTA-ℓ₁) indeed worsens ΔF1 and feature absorption despite improving NMSE and dead-latent rates. This is an important edge case, and we have added a preliminary analysis of this phenomenon in the third paragraph of Section 5.4 in the revised manuscript. Crucially, in the TopK setting, transitioning from full amortization to our non-amortized baseline does not merely reduce amortization; it simultaneously changes the mechanism of  sparsity. The dictionary is trained under a hard TopK constraint with a globally shared encoder, whereas the non-amortized codes are obtained via an ℓ₁-based ISTA solver with no parameter sharing across samples. This shift therefore introduces two concurrent changes: a reduction in amortization, and replacement of the hard K-sparse geometry with a soft-thresholded ℓ₁ prior. This mismatch between the training-time hard TopK constraint and the test-time ℓ₁ solver, compounded by the absence of cross-sample parameter sharing, can disrupt the semantic consistency of rare features and thereby exacerbate both feature absorption and feature splitting. Consequently, this anomalous behavior is largely absent in other SAE variants that do not rely on a hard TopK constraint.

---

### Official Review · Reviewer_KjC9 · 2025-10-30

**Soundness:** 2
**Presentation:** 2
**Contribution:** 2
**Rating:** 4
**Confidence:** 3

**Summary:**

The paper argues that amortized inference, central to SAE, conflicts with mechanistic interpretability goals. Amortization trades instance-level optimality for global efficiency, creating an amortization gap. This gap causes pathological effects that degrade monosemantic feature discovery. Through theoretical and empirical analysis, the authors show that increasing sparsity worsens these issues and that optimizing reconstruction-sparsity tradeoffs doesn’t improve interpretability. They test semi- and non-amortized methods, finding that these reduce reconstruction errors, dead latents, and improve interpretability and control. The conclusion urges a shift away from full amortization toward hybrid or per-instance inference.

**Strengths:**

1. Clear articulation of the conceptual tension between global efficiency and instance-level fidelity

2. Novel experimental demonstration of semi-/non-amortized methods that improve feature interpretability.

3. Situates itself well in current mechanistic interpretability discourse (SAEBench, monosemantic features).

**Weaknesses:**

1. The paper argues strongly against amortization but doesn’t fully explore potential middle grounds such as localized amortization, mixture-of-experts encoders, or hierarchical amortization. This makes the recommendation (“reduce investment in amortization”) feel a bit absolute.

2. The broader meaning of “instance-level optimality” for interpretability is underexplored. For example, how does this affect feature alignment across runs, or practical model auditing? The paper hints at these but doesn’t go deep.

3. While the paper demonstrates empirical correlations (e.g., lower amortization gap -> lower NMSE, fewer dead latents), it doesn’t provide causal interventions or ablations proving amortization itself causes the pathologies, rather than merely correlating with them.

**Questions:**

1. Can amortization be regularized rather than abandoned, e.g., local amortization?

2. How does the amortization gap quantitatively relate to interpretability scores beyond NMSE and sparsity metrics?

3. Could amortization’s failure modes differ across modalities (e.g., vision vs language)?

---

> ### Author Response · Authors · 2025-11-21
>
> ## Response to W1:
> We acknowledge the reviewer’s comment regarding the lack of discussion on potential intermediate schemes. In the revised manuscript, we have incorporated two additional SAE variants, BatchTopK SAE and Matryoshka SAE as baseline methods in Section 5, based on local and hierarchical amortization respectively. Corresponding analysis and discussion have been supplemented, with detailed experimental results provided in Table 5 of Appendix G.2.
>
> The results indicate that these two schemes help alleviate issues such as the Dense Rate, feature splitting, and feature absorption. However, they exhibit a high Dead Rate, suggesting relatively low utilization of features. In contrast, the semi-amortized and non-amortized versions of BatchTopK SAE and Matryoshka SAE show significant improvement in both RMSE and Dead Rate, while still maintaining competitive performance in mitigating feature splitting and feature absorption.
>
> The inclusion of these two baseline groups demonstrates that it is possible to achieve considerable performance gains through architectural refinements, without sacrificing the computational efficiency inherent to amortized inference. Accordingly, we have modified the wording in the manuscript that may have previously appeared overly absolute.
> | Model | SAE Variants | Pattern | NMSE | Dead Rate | Dense Rate@0.1 | Dense Rate@0.2 | F1@1 | F1@2 | $\Delta F1$ | Abs. Rate |
> | :--- | :--- | :--- | :--- | :--- | :--- | :--- | :--- | :--- | :--- | :--- |
> | **Pythia-160m**<br>(Layer 8) | **BatchTopK** | Full-Amortized | 0.537 | 0.316 | 0.053 | 0.040 | 0.700 | 0.748 | 0.047 | 0.171 |
> | | | Semi-Amortized | 0.101 | 0.000 | 0.055 | 0.045 | 0.687 | 0.755 | 0.031 | 0.121 |
> | | | Non-Amortized | 0.017 | 0.000 | 0.053 | 0.043 | 0.726 | 0.758 | 0.031 | 0.033 |
> | | **Matryoshka** | Full-Amortized | 0.369 | 0.219 | 0.107 | 0.047 | 0.689 | 0.722 | 0.033 | 0.240 |
> | | | Semi-Amortized | 0.033 | 0.000 | 0.121 | 0.055 | 0.681 | 0.716 | 0.035 | 0.219 |
> | | | Non-Amortized | 0.011 | 0.007 | 0.114 | 0.049 | 0.677 | 0.729 | 0.052 | 0.172 |
> | | **LocA-SAE** | Loc-Amortized | 0.427 | 0.000 | 0.079 | 0.044 | 0.714 | 0.728 | 0.013 | 0.055 |
> | **Gemma-2-2b**<br>(Layer 12) | **BatchTopK** | Full-Amortized | 0.152 | 0.136 | 0.231 | 0.147 | 0.646 | 0.708 | 0.062 | 0.093 |
> | | | Semi-Amortized | 0.021 | 0.000 | 0.230 | 0.150 | 0.648 | 0.682 | 0.034 | 0.003 |
> | | | Non-Amortized | 0.005 | 0.000 | 0.235 | 0.154 | 0.629 | 0.669 | 0.040 | 0.018 |
> | | **Matryoshka** | Full-Amortized | 0.573 | 0.113 | 0.077 | 0.037 | 0.690 | 0.681 | -0.008 | 0.732 |
> | | | Semi-Amortized | 0.059 | 0.000 | 0.083 | 0.043 | 0.685 | 0.678 | -0.006 | 0.431 |
> | | | Non-Amortized | 0.012 | 0.000 | 0.083 | 0.040 | 0.613 | 0.654 | 0.041 | 0.020 |
> | | **LocA-SAE** | Loc-Amortized | 0.211 | 0.000 | 0.017 | 0.066 | 0.670 | 0.694 | 0.024 | 0.023 |
> ## Response to W2:
> We appreciate the reviewer for pointing this out, which is a valuable suggestion. By "instance-level optimality'' we mean the regime where, for each activation $x$, the amortized code $z_a = f_\phi(x)$ is close to the per-example optimum $z_o(x)$ of the sparse coding objective. This is stronger than achieving a good \emph{average} reconstruction--sparsity trade-off: it requires that every token is encoded by (approximately) the same sparse solution that an ideal per-sample optimizer would produce, and monosemanticity is then defined at this instance-wise level.
> This has direct implications for feature alignment across runs. Fully amortized SAEs, trained only for global expected reconstruction under a fixed sparsity budget, are incentivized to keep high-frequency, cross-domain reusable directions and to absorb or split rarer concepts, which makes features sensitive to initialization and data shifts and hence hard to align or ``stitch'' across runs. In contrast, the instance-wise optimum $z_o(x)$ depends only on the dictionary and sparsity penalty, and our semi-/non-amortized methods move $z_a$ closer to $z_o$, systematically reducing absorption, splitting, and dense latents. We will add a short paragraph in Section 4.2 explicitly making this connection and positioning our results as a mechanistic explanation of why current amortized SAEs struggle to find canonical units.
> For practical auditing, many workflows require (i) reliably localizing the representation of a given concept and (ii) intervening on it with minimal off-target effects. Our TPP and GIS experiments are exactly auditing-style proxies: we use probes or generative interventions to test how selectively concept removal/editing can be achieved. We observe that reducing amortization improves the selectivity of these interventions while keeping the decoder and sparsity level comparable.

---

> ### Author Response · Authors · 2025-11-21
>
> ## Response to W3:
> The purpose of this paper is not to establish a relationship between the reduction of the amortization gap and monosemanticity. On the contrary, in Section 4.1 of the manuscript, we characterize the Amortization Gap as a form of global Pareto improvement in reconstruction and sparsity, which is also the objective pursued by most existing variants of SAEs. Nevertheless, while global Pareto improvement is evaluated at the dataset level, monosemanticity emphasizes instance-level optimality. Consequently, even when both dense latents and dead latents coexist within a set of samples, the Amortization Gap may not significantly differ compared to another set characterized by monosemantic features. To substantiate this claim, we conducted experiments on training dynamics in Section 4.3, tracking the evolution of various pathological metrics and the Amortization Gap across different training steps. The experimental results corroborate our hypothesis: although the Amortization Gap exhibits a declining trend as training progresses, issues such as the Dead Rate, Feature Splitting, and Feature Absorption do not demonstrate a corresponding decrease.
>
> We acknowledge the reviewer’s concern regarding the lack of causal intervention and ablation experiments, which limits the persuasiveness of our claims. In Section 5 of the paper, we introduced semi-amortized and non-amortized setups precisely to observe how various pathological metrics change as dependence on full amortization is reduced. This approach effectively serves as a form of intervention and ablation, and the results demonstrate that reducing reliance on full amortization alleviates multiple pathological phenomena, such as RMSE and Dead Rate. In the revised version of the manuscript (Section 5.3), we explicitly frame the original semi-amortized and non-amortized configurations as intervention and ablation studies, and further propose a locally amortized variant, named LocA-SAE, as a novel methodological contribution.
>
> ## Response to Q1:
> We appreciate the reviewers' constructive feedback. Admittedly, per-sample optimization worsens the inherent scalability issues of SAE, which is not practicable when dealing with large samples. In the revised manuscript, we treat semi-amortized and non-amortized inference as interventions and ablation studies (Sec. 5.3). Furthermore, in Section 5.1 we introduce **LocA-SAE** (Locally Amortized Sparse Autoencoder), a practical SAE variant based on local amortization.
>
> LocA-SAE retains a single globally shared dictionary but partitions the latent variables into groups according to the angular variance of their activation directions across the dataset. Latents exhibiting similar degrees of polysemanticity are assigned to the same group; each group is then equipped with its own dedicated lightweight encoder and receives an independent sparsity penalty. Angular variance serves as a proxy for the diversity of activation patterns supported by a given latent and thus indirectly measures its degree of polysemanticity. By grouping latents with comparable polysemanticity and applying group-specific encoding and sparsity regularization, LocA-SAE largely eliminates the unfair competition that arises under global reconstruction–sparsity constraints when latents of widely differing polysemanticity are forced to share the same encoder and regularization strength.
>
> The detailed methodology and corresponding experimental results have been incorporated into Section 5 of the revised manuscript. Preliminary results demonstrate that LocA-SAE substantially mitigates pathological phenomena such as feature splitting and feature absorption with only a minor degradation in reconstruction performance, and even completely eliminates dead latent in all cases.
> | Model | SAE Variants | Pattern | NMSE | Dead Rate | Dense Rate@0.1 | Dense Rate@0.2 | F1@1 | F1@2 | $\Delta F1$ | Abs. Rate |
> | :--- | :--- | :--- | :--- | :--- | :--- | :--- | :--- | :--- | :--- | :--- |
> | **Pythia-160m**<br>(Layer 8) | **LocA-SAE** | Loc-Amortized | 0.427 | 0.000 | 0.079 | 0.044 | 0.714 | 0.728 | 0.013 | 0.055 |
> | **Gemma-2-2b**<br>(Layer 12) | **LocA-SAE** | Loc-Amortized | 0.211 | 0.000 | 0.017 | 0.066 | 0.670 | 0.694 | 0.024 | 0.023 |

---

> ### Author Response · Authors · 2025-11-21
>
> ## Response to Q2:
>
> In Section 4.1 of the manuscript, we characterize the Amortization Gap as a form of global Pareto improvement in reconstruction and sparsity—precisely the objective pursued by most existing variants of SAEs. Nevertheless, while global Pareto improvement is evaluated at the dataset level, monosemanticity emphasizes instance-level optimality. Consequently, even when both dense latents and dead latents coexist within a set of samples, the Amortization Gap may not significantly differ compared to another set characterized by monosemantic features. To substantiate this claim, we conducted experiments on training dynamics in Section 4.3, tracking the evolution of various pathological metrics and the Amortization Gap across different training steps. The experimental results corroborate our hypothesis: although the Amortization Gap exhibits a declining trend as training progresses, issues such as the Dead Rate, Feature Splitting, and Feature Absorption do not demonstrate a corresponding decrease.
>
> In summary, one of the contribution of our paper is to demonstrate that reducing the Amortization Gap does not necessarily lead to improved monosemanticity, the Amortization Gap and interpretability scores (or degree of monosemanticity) are not strongly correlated.
> ## Response to Q3:
> We acknowledge the reviewer's valid concern regarding the variability of amortization failure modes across different modalities. It is highly probable that these failure modes are intrinsically linked to the underlying modality, given that vision and language models exhibit distinct differences in inductive biases, architectural designs, and sparsity structures. However, our current work deliberately focuses on language models, as the interplay between amortized inference and the specific interpretability pathologies we investigate is already sufficiently intricate and computationally demanding to characterize systematically even within this single modality. Conducting a rigorous cross-modal study would necessitate the design of tailored SAE architectures and evaluation protocols specific to vision. We consider this a substantial undertaking that falls beyond the scope of the current paper, though it certainly merits further exploration as a future direction.

---

> > ### Comment · Reviewer_KjC9 · 2025-11-24
> >
> > Thanks so much for the detailed responses and the additional experiments. Most of my concerns have been addressed. With the proposed changes, I believe the paper provides a very clear message to the community. I have raised my score accordingly.

---

### Official Review · Reviewer_S8RY · 2025-11-01

**Soundness:** 3
**Presentation:** 2
**Contribution:** 2
**Rating:** 4
**Confidence:** 2

**Summary:**

This paper argues that the amortized inference framework, standard in Sparse Autoencoders (SAEs), is fundamentally misaligned with the goal of discovering monosemantic features for mechanistic interpretability. The authors state that the global optimization objective of a shared encoder inherently conflicts with the need for instance-level optimality, leading to pathological behaviors like feature splitting and absorption. They provide empirical evidence showing that moving towards instance-specific, non-amortized optimization significantly improves reconstruction and mitigates these issues, albeit at a high computational cost. Based on this, the authors advocate for a paradigm shift away from purely amortized methods.

**Strengths:**

1. The paper clearly articulates a critical, and perhaps under-appreciated, conflict in the SAE literature: the trade-off between the computational efficiency of amortized inference and the instance-level fidelity required for true monosemanticity. This reframing of common SAE pathologies as symptoms of an inference problem, rather than just an architectural one, is a valuable conceptual contribution.

2. The core experiment comparing fully-amortized, semi-amortized, and non-amortized inference using the same trained decoder is convincing. It effectively isolates the inference strategy as the key variable, providing direct and convincing evidence that the amortization gap is a significant source of suboptimality and pathological behavior in SAEs

**Weaknesses:**

1. The paper's main weakness is that it diagnoses a problem without offering a viable solution. The proposed non-amortized alternative, which relies on per-sample iterative optimization (e.g., 200 ISTA steps), is computationally intractable for the very large-scale applications where SAEs are most needed. To make the call for a paradigm shift convincing, the authors should propose and evaluate at least one hybrid approach that balances instance-level fidelity with computational feasibility.

2. The central concept, the "amortization gap," is a well-established phenomenon extensively studied in the context of Variational Autoencoders and recent work on SAEs (e.g., O'Neill et al. 2024), which substantially detracts from the informativeness of this work. It would be great if the authors could make the ``delta'' more transparent.

**Questions:**

Please see the weaknesses section.

In addition, the non-amortized approach appears to worsen some pathologies (feature splitting and absorption) for TopK SAEs on Gemma-2-2b. How do you reconcile this with your central thesis that amortization is the primary cause of these problems?

---

> ### Author Response · Authors · 2025-11-21
>
> ## Response to W1:
> We acknowledge the reviewer’s concern regarding the prohibitive computational cost of ISTA-based non-amortized baselines. In the revised manuscript, we treat semi-amortized and non-amortized inference as interventions and ablation studies (Sec. 5.3). Furthermore, in Section 5.1 we introduce **LocA-SAE** (Locally Amortized Sparse Autoencoder), a practical SAE variant based on local amortization.
>
> LocA-SAE retains a single globally shared dictionary but partitions the latent variables into groups according to the angular variance of their activation directions across the dataset. Latents exhibiting similar degrees of polysemanticity are assigned to the same group; each group is then equipped with its own dedicated lightweight encoder and receives an independent sparsity penalty. Angular variance serves as a proxy for the diversity of activation patterns supported by a given latent and thus indirectly measures its degree of polysemanticity. By grouping latents with comparable polysemanticity and applying group-specific encoding and sparsity regularization, LocA-SAE largely eliminates the unfair competition that arises under global reconstruction–sparsity constraints when latents of widely differing polysemanticity are forced to share the same encoder and regularization strength.
>
> The detailed methodology and corresponding experimental results have been incorporated into Section 5 of the revised manuscript. Preliminary results demonstrate that LocA-SAE substantially mitigates pathological phenomena such as feature splitting and feature absorption with only a minor degradation in reconstruction performance, and even completely eliminates dead latent in all cases.
> | Model | SAE Variants | Pattern | NMSE | Dead Rate | Dense Rate@0.1 | Dense Rate@0.2 | F1@1 | F1@2 | $\Delta F1$ | Abs. Rate |
> | :--- | :--- | :--- | :--- | :--- | :--- | :--- | :--- | :--- | :--- | :--- |
> | **Pythia-160m**<br>(Layer 8) | **BatchTopK** | Full-Amortized | 0.537 | 0.316 | 0.053 | 0.040 | 0.700 | 0.748 | 0.047 | 0.171 |
> | | | Semi-Amortized | 0.101 | 0.000 | 0.055 | 0.045 | 0.687 | 0.755 | 0.031 | 0.121 |
> | | | Non-Amortized | 0.017 | 0.000 | 0.053 | 0.043 | 0.726 | 0.758 | 0.031 | 0.033 |
> | | **Matryoshka** | Full-Amortized | 0.369 | 0.219 | 0.107 | 0.047 | 0.689 | 0.722 | 0.033 | 0.240 |
> | | | Semi-Amortized | 0.033 | 0.000 | 0.121 | 0.055 | 0.681 | 0.716 | 0.035 | 0.219 |
> | | | Non-Amortized | 0.011 | 0.007 | 0.114 | 0.049 | 0.677 | 0.729 | 0.052 | 0.172 |
> | | **LocA-SAE** | Loc-Amortized | 0.427 | 0.000 | 0.079 | 0.044 | 0.714 | 0.728 | 0.013 | 0.055 |
> | **Gemma-2-2b**<br>(Layer 12) | **BatchTopK** | Full-Amortized | 0.152 | 0.136 | 0.231 | 0.147 | 0.646 | 0.708 | 0.062 | 0.093 |
> | | | Semi-Amortized | 0.021 | 0.000 | 0.230 | 0.150 | 0.648 | 0.682 | 0.034 | 0.003 |
> | | | Non-Amortized | 0.005 | 0.000 | 0.235 | 0.154 | 0.629 | 0.669 | 0.040 | 0.018 |
> | | **Matryoshka** | Full-Amortized | 0.573 | 0.113 | 0.077 | 0.037 | 0.690 | 0.681 | -0.008 | 0.732 |
> | | | Semi-Amortized | 0.059 | 0.000 | 0.083 | 0.043 | 0.685 | 0.678 | -0.006 | 0.431 |
> | | | Non-Amortized | 0.012 | 0.000 | 0.083 | 0.040 | 0.613 | 0.654 | 0.041 | 0.020 |
> | | **LocA-SAE** | Loc-Amortized | 0.211 | 0.000 | 0.017 | 0.066 | 0.670 | 0.694 | 0.024 | 0.023 |
>
> ## Response to W2:
> We acknowledge that the Amortization Gap is already a well-studied phenomenon in both the VAE and SAE literature. However, the contribution of this work lies in revealing that the Amortization Gap obscures inherent trade-offs among various pathological phenomena. In Section 4.1 of the manuscript, we characterize the Amortization Gap as a form of global Pareto improvement in reconstruction and sparsity, which is also the objective pursued by most existing variants of SAEs. Nevertheless, while global Pareto improvement is evaluated at the dataset level, monosemanticity emphasizes instance-level optimality. Consequently, even when both dense latents and dead latents coexist within a set of samples, the Amortization Gap may not significantly differ compared to another set characterized by monosemantic features. To substantiate this claim, we conducted experiments on training dynamics in Section 4.3, tracking the evolution of various pathological metrics and the Amortization Gap across different training steps. The experimental results corroborate our hypothesis: although the Amortization Gap exhibits a declining trend as training progresses, issues such as the Dead Rate, Feature Splitting, and Feature Absorption do not demonstrate a corresponding decrease.

---

> ### Author Response · Authors · 2025-11-22
>
> ## Response to Question:
> We appreciate the reviewer’s careful reading and agree that, for TopK SAEs trained on Gemma-2-2b, our non-amortized baseline (ISTA-ℓ₁) indeed worsens ΔF1 and feature absorption despite improving NMSE and dead-latent rates. This is an important edge case, and we have added a preliminary analysis of this phenomenon in the third paragraph of Section 5.4 in the revised manuscript. Crucially, in the TopK setting, transitioning from full amortization to our non-amortized baseline does not merely reduce amortization; it simultaneously changes the mechanism of  sparsity. The dictionary is trained under a hard TopK constraint with a globally shared encoder, whereas the non-amortized codes are obtained via an ℓ₁-based ISTA solver with no parameter sharing across samples. This shift therefore introduces two concurrent changes: a reduction in amortization, and replacement of the hard K-sparse geometry with a soft-thresholded ℓ₁ prior. This mismatch between the training-time hard TopK constraint and the test-time ℓ₁ solver, compounded by the absence of cross-sample parameter sharing, can disrupt the semantic consistency of rare features and thereby exacerbate both feature absorption and feature splitting. Consequently, this anomalous behavior is largely absent in other SAE variants that do not rely on a hard TopK constraint.

---

### Official Review · Reviewer_Ycij · 2025-11-02

**Soundness:** 3
**Presentation:** 3
**Contribution:** 3
**Rating:** 6
**Confidence:** 4

**Summary:**

The paper considers the potentially overlooked interrelation between mono-semanticity and reconstruction-sparsity tradeoff in the SAE context. Specifically, the (position) paper presents empirical findings including (1) improvement over the reconstruction-sparsity Pareto frontier comes at the expense of dictionary capacity and mono-semanticity, and (2) semi-/non- amortized encoding approaches show benefits in improving reconstruction and features (e.g., concept removal tasks) and in addressing dead latents issue. The paper argues that the conflict between parameter-sharing encoding and instance-level optimality stems from the SAE architecture itself, and cautions against the investment in amortization-based encoding for polysemy disentanglement.

**Strengths:**

### Writing

Overall, the paper is relatively easy to follow. The problem setting of interest, the previous approaches and how they fit in the discussion, the purpose of empirical evaluations, the settings of experiments and findings are presented in an organized and clear way.

### Significance

The paper (or position paper?) aims to present a case/caution against the investment in amortization-based encoding, if the goal includes polysemy disentanglement. This is carried out relatively well, since the arguments are well-structured along with empirical evidence.

**Weaknesses:**

### Novelty

The paper strikes me more of a position paper than a regular conference paper (just stating this as a neutral comment). There are potential some concerns on the novelty (detailed in "Questions" section).

### Minor Things

- please fix the opening quotation marks in lines 160, 192, 243

**Questions:**

- Question/Comment on novelty

The paper lays out issues ("pathological phenomena") identified in previous works, e.g., dead latents (Gao et al., 2024), dense latents (Sun et al., 2025), feature splitting and feature absorption (Chanin et al. 2024), and also reproduces some exps from Karvonen et al. (2025). It would be very helpful if the authors can be more explicit about the unique contribution to address the potential concern on novelty.

---

> ### Author Response · Authors · 2025-11-21
>
> ## Response to Question:
> We acknowledge the reviewers' concerns regarding novelty. In the revised manuscript, we fix the the opening quotation issues, and treat semi-amortization and non-amortization as intervention and ablation experiments, respectively, to validate the relationship between the degree of reliance on amortization inferences and pathological indicators (Section 5.3). Furthermore, in Section 5.1 we introduce LocA-SAE (Locally Amortized Sparse Autoencoder), a practical SAE variant based on local amortization. LocA-SAE retains a single globally shared dictionary but partitions the latent variables into groups according to the angular variance of their activation directions across the dataset. Latents exhibiting similar degrees of polysemanticity are assigned to the same group; each group is then equipped with its own dedicated lightweight encoder and receives an independent sparsity penalty. Angular variance serves as a proxy for the diversity of activation patterns supported by a given latent and thus indirectly measures its degree of polysemanticity. By grouping latents with comparable polysemanticity and applying group-specific encoding and sparsity regularization, LocA-SAE largely eliminates the unfair competition that arises under global reconstruction–sparsity constraints when latents of widely differing polysemanticity are forced to share the same encoder and regularization strength.
>
> The detailed methodology and corresponding experimental results have been incorporated into Section 5 of the revised manuscript. Preliminary results demonstrate that LocA-SAE substantially mitigates pathological phenomena such as feature splitting and feature absorption with only a minor degradation in reconstruction performance, and even completely eliminates dead latent in all cases.
>
> | Model | SAE Variants | Pattern | NMSE | Dead Rate | Dense Rate@0.1 | Dense Rate@0.2 | F1@1 | F1@2 | $\Delta F1$ | Abs. Rate |
> | :--- | :--- | :--- | :--- | :--- | :--- | :--- | :--- | :--- | :--- | :--- |
> | **Pythia-160m**<br>(Layer 8) | **BatchTopK** | Full-Amortized | 0.537 | 0.316 | 0.053 | 0.040 | 0.700 | 0.748 | 0.047 | 0.171 |
> | | | Semi-Amortized | 0.101 | 0.000 | 0.055 | 0.045 | 0.687 | 0.755 | 0.031 | 0.121 |
> | | | Non-Amortized | 0.017 | 0.000 | 0.053 | 0.043 | 0.726 | 0.758 | 0.031 | 0.033 |
> | | **Matryoshka** | Full-Amortized | 0.369 | 0.219 | 0.107 | 0.047 | 0.689 | 0.722 | 0.033 | 0.240 |
> | | | Semi-Amortized | 0.033 | 0.000 | 0.121 | 0.055 | 0.681 | 0.716 | 0.035 | 0.219 |
> | | | Non-Amortized | 0.011 | 0.007 | 0.114 | 0.049 | 0.677 | 0.729 | 0.052 | 0.172 |
> | | **LocA-SAE** | Loc-Amortized | 0.427 | 0.000 | 0.079 | 0.044 | 0.714 | 0.728 | 0.013 | 0.055 |
> | **Gemma-2-2b**<br>(Layer 12) | **BatchTopK** | Full-Amortized | 0.152 | 0.136 | 0.231 | 0.147 | 0.646 | 0.708 | 0.062 | 0.093 |
> | | | Semi-Amortized | 0.021 | 0.000 | 0.230 | 0.150 | 0.648 | 0.682 | 0.034 | 0.003 |
> | | | Non-Amortized | 0.005 | 0.000 | 0.235 | 0.154 | 0.629 | 0.669 | 0.040 | 0.018 |
> | | **Matryoshka** | Full-Amortized | 0.573 | 0.113 | 0.077 | 0.037 | 0.690 | 0.681 | -0.008 | 0.732 |
> | | | Semi-Amortized | 0.059 | 0.000 | 0.083 | 0.043 | 0.685 | 0.678 | -0.006 | 0.431 |
> | | | Non-Amortized | 0.012 | 0.000 | 0.083 | 0.040 | 0.613 | 0.654 | 0.041 | 0.020 |
> | | **LocA-SAE** | Loc-Amortized | 0.211 | 0.000 | 0.017 | 0.066 | 0.670 | 0.694 | 0.024 | 0.023 |

---

### Author Response · Authors · 2025-11-24
**Follow-up on Reviewer Feedback for Paper #8846**

Dear Area Chair,

We are authors of the ICLR 2026 submission #8846, titled "The Price of Amortized Inference in Sparse Autoencoders". We have submitted our complete rebuttal and additional experimental results on Nov. 22 to address the reviewers' valuable comments. We would like to politely inquire if there have been any further updates or feedback from the reviewers, as we have not yet seen any responses on OpenReview.

We greatly appreciate the reviewers' insights, which have helped us strengthen our work. We are more than willing to provide further clarification should any questions remain from our rebuttal.

Thank you for your time and consideration.

Sincerely,

Anonymous Author

Submission #8846

---

> ### Comment · Reviewer_KjC9 · 2025-11-24
>
> Dear Authors,
>
> Thanks for your responses. I have seen your updated PDF. However, it seems that your rebuttals on OpenReview are not visible to me. I'm not sure if it is an issue from my end. Could you please check whether Reviewers are included as readers?
>
> Best,
>
> Reviewer KjC9

---

> > ### Author Response · Authors · 2025-11-24
> >
> > Dear Reviewer,
> >
> > We have just updated the visibility status of the rebuttal, apologize for any inconvenience caused. Looking forward to your reply.
> >
> > Sincerely,
> >
> > Authors of Submission #8846

---

### Author Response · Authors · 2025-11-25
**Follow-up Inquiry: Rebuttal Status for Submission #8846**

Dear Area Chair,

I am writing as a follow-up regarding our ICLR 2026 submission #8846, "The Price of Amortized Inference in Sparse Autoencoders".

We sincerely appreciate the initial engagement from one of the reviewers. However, with the rebuttal period concluding, we are concerned that we have not yet received feedback from the other three reviewers. Our rebuttal included significant additional experiments and detailed point-by-point responses to all raised concerns.

We are very eager to ensure that all reviewers have the opportunity to consider our responses, as their insights are crucial for a fair evaluation. Thank you for your attention and support in this matter.

Sincerely,

Authors of Submission #8846

---

### Author Response · Authors · 2025-11-28
**Polite Follow-up on Rebuttal Status for Submission #8846**

Dear Area Chair,

I am writing to follow up once more on our ICLR 2026 submission #8846, "The Price of Amortized Inference in Sparse Autoencoders".

Thus far, we have only received follow-up feedback from one of the four assigned reviewers. We wanted to kindly bring this to your attention, as we are eager to ensure that our detailed rebuttal and new experimental results are considered by the full committee before final decisions are made.

We understand the considerable demands on the reviewers' time. However, given the circumstances, we would be truly grateful for any guidance you might offer or if it might be possible to facilitate a final check-in with the reviewers.

Thank you so much for your time and for your stewardship of the review process.

Sincerely,

Authors of Submission #8846

---

### Author Response · Authors · 2025-11-29
**Summary of Revisions to Address Reviewers' Concerns and Questions**

We have uploaded a revised manuscript that directly addresses the reviewers’ three main concerns; major changes are highlighted in the PDF.

1. **Practical Alternative: LocA-SAE**
   Reviewers pointed out that although we established a connection between amortized inference and the pathological behaviors of SAEs, the proposed semi-amortized and non-amortized approaches are too computationally expensive and thus lack a practical alternative solution. In response, we propose **LocA-SAE**, which keeps a single global dictionary but groups latents by the angular variance of their activations and assigns each group a lightweight encoder and sparsity penalty. This locally amortized design substantially reduces feature splitting, and absorption, and even eradicated dead latent, with only minor reconstruction degradation.

2. **Intervention/Ablation View and Richer Baselines**
   Concerns about “only correlations” and missing intermediate schemes are addressed by making explicit that our semi- and non-amortized setups already act as interventions on the degree of amortization:  the "semi-amortized" variant uses the original SAE encoder as initialization and refines codes with a limited number of ISTA steps and the "non-amortized" variant relies purely on ISTA without using the encoder.
   We now present this as a controlled knob on amortization strength in Section 5.3, and we add BatchTopK SAE and Matryoshka SAE as local and hierarchical amortization baselines, each evaluated in full-, semi-, and non-amortized forms.

3. **TopK–Gemma Anomaly: Added Explanation**
   For the anomalous case where **TopK SAEs on Gemma-2-2b** show worse ΔF1 and feature absorption under non-amortized ISTA–ℓ₁ despite better NMSE and dead-latent rates, we add a dedicated discussion in Section 5.4. We explain that in this setting, moving to the non-amortized baseline both reduces amortization *and* switches from a hard TopK training constraint to a soft ℓ₁ prior at test time without parameter sharing, and we argue that this mismatch can destabilize rare features—an effect largely absent in non-TopK SAE variants.

After these revisions and the author–reviewer discussion, the scores improved from **6, 4, 4, 6** to **6, 4, 6, 6**, with Reviewer KjC9 explicitly noting that most concerns were resolved and raising their score.

---

### Meta-Review · Area_Chair_x3ZG · 2026-01-08

**Summary:**

This study examines a core tension in sparse autoencoders for mechanistic interpretability: sharing a single amortized encoder can optimise reconstruction and sparsity at the dataset level while undermining the instance-level optimality needed for stable, monosemantic and “stitchable” features. Through analyses of training dynamics and controlled comparisons between fully amortized, semi-amortized  and non-amortized inference, the authors show that increased sparsity can amplify multiple pathological behaviours—including dead and dense latents, feature splitting and feature absorption—and argue that these trade-offs are driven by amortization.

In response to reviewer feedback, the revised manuscript strengthens both the framing and the practical relevance by treating semi-/non-amortized settings explicitly as interventions on amortization strength, expanding baselines (including BatchTopK and Matryoshka variants), and proposing LocA-SAE as a more feasible hybrid alternative that reduces key pathologies while largely preserving reconstruction fidelity.

**Reviewer Concerns:**

The rebuttal substantially resolves the principal objections around practicality and experimental completeness by introducing LocA-SAE as a computationally plausible compromise, broadening the comparison set with additional amortization-aware baselines and evaluating them under full-, semi- and non-amortized inference, clarifying the intervention logic underlying the ablations, and adding targeted discussion for the TopK–Gemma anomaly alongside a more systematic exploration of semi-amortized refinement budgets. Outstanding concerns are narrower but still material: the central link between instance-level optimality and monosemanticity remains supported mainly by empirical proxies rather than a fully convincing mechanistic account, the scope of evidence is still largely confined to the studied language-model settings (leaving questions about generality across modalities and regimes), and the manuscript would benefit from an even clearer delineation of what is genuinely new relative to established “amortization gap” discussions in the VAE/SAE literature.

**Reviewer Scores:**

Reviewer assessments began mixed but borderline overall, with scores spanning marginally below to marginally above threshold (6, 4, 4, 6), reflecting shared interest in the framing but uneven confidence in novelty and actionable outcomes. After rebuttal and revision, at least one reviewer explicitly increased their score and the aggregate shifted upward (6, 4, 6, 6).

---

### Decision · Program_Chairs · 2026-01-26

Accept (Poster)